# Rethinking the Influence of Distribution Adjustment in Incremental Segmentation

## Abstract

In an ever-changing world, incremental segmentation learning faces challenges due to the need for pixel-level accuracy and the practical application of gradually obtained samples. While most existing methods excel in stability by freezing model parameters or employing other regularization techniques to preserve the distribution of old knowledge, these approaches often fall short of achieving satisfactory plasticity. This phenomenon arises from the limited allocation of parameters for learning new knowledge. Meanwhile, in such a learning manner, the distribution of old knowledge cannot be optimized as new knowledge accumulates. As a result, the feature distribution of newly learned knowledge overlaps with old knowledge, leading to inaccurate segmentation performance on new classes and insufficient plasticity. This issue prompts us to explore how both old and new knowledge representations can be dynamically and simultaneously adjusted in the feature space during incremental learning. To address this, we conduct a mathematical structural analysis, which indicates that compressing the feature subspace and promoting sparse distribution is beneficial in allocating more space for new knowledge in incremental segmentation learning. Following compression principles, high-dimensional knowledge is projected into a lower-dimensional space in a contracted and dimensionally reduced manner. Regarding sparsity, the exclusivity of multiple peaks in Gaussian mixture distributions across different classes is preserved. Through effective knowledge transfer, both up-to-date and long-standing knowledge can dynamically adapt within a unified space, facilitating efficient adaptation to continuously incoming and evolving data. Extensive experiments across various incremental settings consistently demonstrate the significant improvements provided by our proposed method. In particular, regarding the plasticity of in the incremental stage, our approach outperforms the state-of-the-art method by $11.7\%$ in MIoU scores for the challenging 10-1 setting. Source code is available in the supplementary materials.

## 1 Introduction

Incremental learning, which mimics the dynamic nature of real-world data acquired progressively, requiring adaptation to all previously encountered data, is widely applicable across various scenarios, such as robot sensing, autonomous driving, and beyond. The primary objective is to acquire current knowledge while retaining long-standing knowledge, without reliance on joint training (Masana et al., 2020). Based on this objective, the stability-plasticity dilemma represents the core challenge that incremental learning aims to overcome. Artificially fixing the parameters of previous learning can ensure high stability (preventing catastrophic forgetting) but it frequently results in inadequate plasticity (constraining the algorithm's ability to acquire new knowledge). While the majority of incremental approaches have concentrated on addressing incremental classification learning, recent developments have broadened incremental learning to more intricate pixel-wise incremental segmentation (Yuan & Zhao, 2023).

Several existing methods (Cha et al., 2021; Zhang et al., 2022b; Yang et al., 2023) for incremental segmentation have endeavored to resolve the stability-plasticity dilemma, achieving notable advancements in terms of performance. Particularly, they have attained stability levels comparable to joint training accuracy. These effective strategies encompass a variety of methodologies, focusing primarily on regularization-based, expanding architecture-based, and memory replay-based tech-

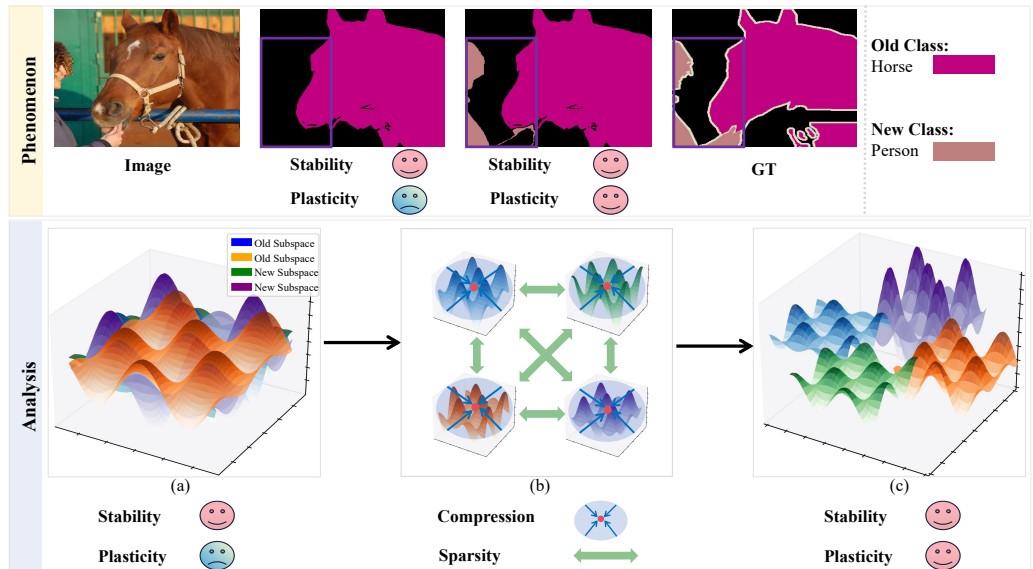

Figure 1: Phenomenon and analysis of good stability and limited plasticity. Maintaining fixed old knowledge results in overlapping subspaces, which hinders the formation of discriminative features and leads to limited plasticity. Effectively reconstructing the feature distributions of both old and new knowledge can promote better plasticity.

niques. While these diverse methods seek to preserve previously learned knowledge with minimal modification, this constraint reduces flexibility for adapting to new knowledge and results in inadequate plasticity. In other words, for these methods, preserving old knowledge in an unchanged state effectively combats catastrophic forgetting, but impairs the ability to assimilate new knowledge.

Whether by freezing a substantial portion of the model (Cha et al., 2021; Zhang et al., 2022b) or by requiring model to optimize itself to the initial state of old knowledge in the incremental stage (Shan et al., 2022; Yang et al., 2022; Wu et al., 2023), these methods induce subtle variations in the information that affect old knowledge. Such learning manners result in overlapping category subspaces during the incremental stage (see Figure 1 (a)), creating difficulties in generating discriminative features for both new information and existing knowledge.

In this regard, we derive insights from the effects of representation distribution among different categories. That is, we can mitigate the constraints imposed by preserving the invariance of old knowledge in incremental segmentation. We endeavor to alleviate the overlap in subspace distribution and promote the formation of more discriminative features. Recent studies (Kim et al., 2024; Wuerkaixi et al., 2024) indicate that dynamically adjusting learned knowledge is effective for domain incremental learning. However, in the field of incremental segmentation, most methods (Gong et al., 2024; Yang et al., 2023) maintain the learned knowledge for good stability. Since it is more challenging to achieve a balance between stability and plasticity using dynamic adjustment methods due to the requirements of pixel-level precision. Allowing variability in subspace distributions for both new and old knowledge leads to loosely coupled subspace distributions, which provide differentiated feature information to maintain the stability and plasticity of the incremental segmentation, as illustrated in Figure 1 (b) and Figure 1 (c).

Motivated by the observed phenomenon that excessive reliance on old knowledge leads to unsatisfied plasticity, we propose a more realistic and challenging learning paradigm in this paper: enabling the dynamic adaptation of parameters that affect knowledge retention, including both general knowledge and class-specific knowledge. From a feature perspective, when encountering the embedding of feature distributions from new categories, maintaining the invariance of old categories often results in inadequate discriminative feature representation, thereby constraining performance improvements, as illustrated in the second row of Figure 2. To tackle this issue, we conduct mathematical analysis and modeling of incremental segmentation, emphasizing the importance of introducing compression and sparsity in the feature space. This factor is critical for balancing stability and plasticity,

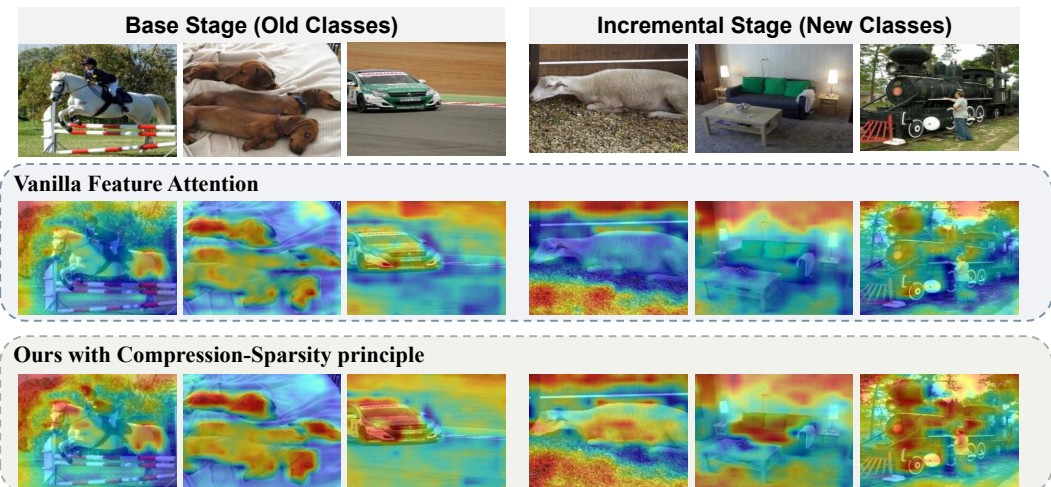

Figure 2: Motivation behind proposed compression-sparsity principle. We visualize the feature attention to illustrate the advantages of our method. In vanilla feature attention (second row), we observe that weaker feature attention responses for different objects , resulting in insufficient discriminative features. In our proposed method (last row), we reveal the causes of this degradation and activate the latent diverse representation.

ultimately enhancing long-term algorithmic performance, as shown in the last row of Figure 2. Our work introduces a practical and innovative approach for modifying feature distributions, referred to **C**ompression-**S**parsity based **I**ncremental **S**egmentation **L**earning (CSISL). Compression is applied to the knowledge structure of complex networks, encompassing both multi-class general knowledge and class-specific knowledge. The knowledge is subsequently mapped into three-dimensional space, with each dimension corresponding to the horizontal position, vertical position, and feature response information. Besides dimensionality reduction, the feature mapping process involves learning optimal compression parameters to narrow the range of feature responses and minimize spatial overlap in the distribution. Sparsity is attained by identifying and constraining the multi-peaks present within the Gaussian mixture distribution in the compressed three-dimensional subspace. The objective is to enhance the separation between peaks that represent distinct subspaces, thereby maximizing the available subspace for incorporating new knowledge. The core of our approach lies in effectively handling the variability of data encountered in incremental learning, aligning with the dynamic and adaptive requirements of practical learning processes.

Unlike the existing diverse and effective methods currently utilized in incremental learning (Schuster et al., 2021; Wang et al., 2022; Menezes et al., 2024), our approach aims to change the immutability of old knowledge in the field of incremental segmentation. Instead, we facilitate the dynamic adaptation of knowledge by modifying the subspace of Gaussian mixture distribution as new class knowledge is acquired. This adaptability empowers the modification of subspaces, enabling the preservation of more distinguish class features while reducing the coupling of subspace distributions. To further validate the motivation and rationality of our method, we present a mathematical analysis is provided to demonstrate the benefit of the compression-sparsity operation in the feature space. Rather than focusing on maximizing distances between class centroids based on similarity (Ferdinand et al., 2022; Xuan et al., 2024), our approach adopts an additional strategy that maximizes the distances between multiple peaks within Gaussian mixture distribution. This stricter constraint compels the network to more effectively minimize the coupling among different class knowledge distributions, promoting more enhanced and concentrated feature responses. To provide a more intuitive understanding of the enhancement facilitated by the compression-sparsity principle, we conduct experiments in complex incremental settings. The main contributions of this paper could be summarized as follows:

- Mathematical analysis demonstrates the benefit of compression-sparsity in incremental segmentation learning, emphasizing their interdependent role in maintaining stability and plasticity. Compression primarily shrinks the representation of old knowledge, while sparsity

minimizes the overlap among different subspaces. This foundation facilitates the preservation of discriminative features across multiple knowledge classes.

- Based on the principles of compression and sparsity, this paper presents practical implementation techniques. Compression is achieved by reducing the dimensionality space and shrinking the representation of class knowledge, while sparsity is accomplished by minimizing the coupling among subspaces through distance maximization between multiple peaks in the Gaussian mixture distribution.

- Experiments are conducted across various incremental settings, demonstrating the effectiveness of our proposed approach in overcoming plasticity constraints. In the challenging incremental configuration with 11 steps of 10-1, our method achieved improvements of 11.7% in incremental stage categories and 6.4% in overall categories compared to the previous state-of-the-art approaches.

## 2  RELATED WORK

In this section, we review the previous studies on regularization-based learning, expanding architecture-based learning, and memory replay-based learning. By summarizing and analyzing recent methods, we propose a novel learning manner with dynamic adaptation for both old and new knowledge.

**Regularization-based Learning.** These methods (Han et al., 2023; Kim et al., 2024; Zhao et al., 2023; Jiang et al., 2023) constrain parameter values using various loss functions. Common approaches include knowledge distillation (Hinton et al., 2015), contrastive learning (Lin et al., 2023; Ji et al., 2023), and parameter freezing. AFC (Kang et al., 2022) minimizes the upper bound of the loss function and leverages the importance of individual backbone feature maps for knowledge distillation. This effectively mitigates catastrophic forgetting, even with limited data from previous classes. Semi-FSCIL (Cui et al., 2023) applies the nearest-mean-of-exemplars principle to select unlabeled data and uses knowledge distillation to learn from them, thereby improving class means. RCIL (Zhang et al., 2022a) incorporates a structured re-parameterization mechanism and a knowledge distillation strategy based on spatial and channel dimensions to prevent catastrophic forgetting when accommodating new classes. In addition to the conventional knowledge distillation approach, CD (Arnaudo et al., 2021) introduces contrastive regularization. This technique involves comparing each input with its augmented version (e.g., via flipping and rotations) to minimize discrepancies between the segmentation features produced by both inputs. UCD (Yang et al., 2022) introduces an uncertainty-aware contrastive distillation method that encourages high similarity among pixels of the same class while pulling apart the center distances of pixels from different classes. These contrastive features are extracted from both the frozen old knowledge after previous learning steps and the knowledge of the newly learned class. These well-designed methods effectively maintain consistency between the network's representations in the new incremental stage and previous ones by constraining parameters, features, mapping spaces, and other aspects, thus preventing catastrophic forgetting. Nonetheless, although they provide considerable advantages in preserving stability for old tasks, the immutability of old knowledge frequently results in an imbalance between stability and plasticity when new knowledge is learned.

**Expanding Architecture-based Learning.** These methods (Yoon et al., 2017; Qin et al., 2021) aim to allocate specific parameters to each class, potentially leading to a significant increase in model parameters as the number of learned classes grows. To efficiently select the appropriate experts during testing, EG (Aljundi et al., 2016) calculates the correlation between classes and directs the test samples to the corresponding sub-models. PackNet (Mallya & Lazebnik, 2017) modifies fine-tuning parameters and retraining parameters to assign specific parameters for each class, guiding learning and prediction. Although these methodologies dynamically expand network structures as new knowledge is introduced, enhancing plasticity to some extent, they face the practical challenge of unbounded network expansion in real-world applications.

**Memory Replay-based Learning.** These methods (Zhang et al., 2024; Lin et al., 2023) store a limited quantity of historical data to utilize previous information when learning class data. Advancements in generative models (Shin et al., 2017; Wu et al., 2018), even if the historical data is unavailable, enable the effective use of these stored pool samples to supplement the learning process, even in the absence of historical data. SSUL (Cha et al., 2021) combines historical replay and parameter freezing to prevent performance degradation in model stability. A-GEM (Chaudhry et al.,

2018) aims to improve model robustness in non-stationary environments. It estimates the mean of the gradients by leveraging experience data from the memory pool, reducing gradient variance and enhancing model performance on new classes. MER (Riemer et al., 2018) strengthens gradient alignment through meta-learning and experience replay, enabling better adaptation to learning classes in non-stationary environments. The data replay pool is limited in size, thereby not significantly burdening storage in practical applications. Hence, it is progressively becoming a prevalent auxiliary strategy for achieving incremental learning.

## 3 THEORETICAL ANALYSIS OF COMPRESSION-SPARSITY PRINCIPLE

### 3.1 TASK DEFINITION

Incremental Segmentation simulates the gradual emergence of multiple new classes in real-world scenarios by defining a sequence of learning steps, where each step is denoted as t = 1, ..., T. In each learning step t, a dataset $D_t$ and a non-zero number of classes $C_t$ are involved. A model $F_t$ with parameters $\theta$ is constructed to facilitate the segmentation learning, assigning different classes to each pixel. Typically, this model consists of a feature extractor $G_t^\theta$, and a classifier $H_t^\theta$. Assuming that the classes learned in the previous step t-1 are denoted as $C_{t-1}$, and the classes learned in the current step t are denoted as $C_t$. Consistent with prior studies, all steps generally include a background class $C_u$, which may encompass previously learned or unseen classes. The objective of incremental segmentation is to perform pixel-level segmentation of classes $C_{1:t}$ on input images after completing the learning of the t-th step, even without access to all the data $D_{1:t-1}$ at this stage. Consequently, the predicted result $P_t$ includes the segmentation results corresponding to N categories and their corresponding class labels, represented as $P_t = \{(M_i, C_i) \mid M_i \in \{0, 1\}^{H \times W}, C_i \in C\}$.

### 3.2 MATHEMATICAL ANALYSIS OF COMPRESSION-SPARSITY PRINCIPLE

While current algorithms have made significant progress in achieving stability comparable to joint training, a considerable deficiency in plasticity remains when compared to the ideal state. To analyze this issue, we establish mathematical formulas from a probabilistic perspective. Within this analysis, the optimization of network parameters $\theta$ is reformulated as the problem of maximizing the likelihood of $\theta$ given the data X. This can be accomplished using Bayes' theorem as follows:

$$\log P(\theta|X) = \log P(X|\theta) + \log P(\theta) - \log P(X) \tag{1}$$

Assuming X represents the complete dataset for learning, including the data required for joint training. We can formulate the incremental training process by partitioning the data in X into two subsets, $X_1$ and $X_2$, according to their respective categories. This leads to the following formulation:

$$\log P(\theta|X) = \log P(X_2|\theta) + \log P(\theta|X_1) - \log P(X_2) \tag{2}$$

In this equation, $\log P(\theta|X)$ denotes the posterior probability of joint training on $X_1$ and $X_2$, serving as an upper bound on the performance of incremental distribution learning. $\log P(X_2|\theta)$ represents the negative loss incurred during the learning of the new class $X_2$, while the posterior distribution $\log P(\theta|X_1)$ corresponds to the proportion of knowledge assimilated by the network after learning $X_1$. It is important to note that $X_1$ corresponds to the data learned in step 1, and $X_2$ corresponds to the data learned in step 2. Further step divisions are not explicitly considered here, as this simplification is implemented for analytical convenience. Additionally, $logP(\theta|X_1)$ follows a Gaussian mixture distribution, implying that any complex curve can be approximated by a combination of Gaussian curves.

$$\log P(\theta|X_1) = \sum_{k=1}^{K} w_k g(\theta|X_1, \mu_k, \sigma_k) \tag{3}$$

Here, K denotes the number of components in the Gaussian mixture distribution, while $g(\theta|X_1, \mu_k, \sigma_k)$ represents the Gaussian distribution that satisfies the mean $\mu$ and variance $\sigma$ for the current step. At this point, the optimal parameter $\theta^*$ can be estimated as:

$$\theta^* = argmin\{-\log P(\theta|X_1)\} \tag{4}$$

Based on the Taylor expansion, the right-hand side of Equation (3) can be approximated as:

$$\sum_{k=1}^{K} w_k g(\theta|X_1, \mu_k, \sigma_k) \approx -\frac{1}{2}(\theta - \theta^*)^T H(\theta^*)(\theta - \theta^*) + constant \tag{5}$$

Figure 3: Diagram of compression-sparsity based algorithm. This figure illustrates how a dynamically adaptive strategy compacts and sparses knowledge when learning new categories, ensuring the preservation of essential features. Knowledge transfer is utilized to obtain the feature distribution of old categories, facilitating the separation of the peaks of the Gaussian mixture distribution.

where $H(\theta^*)$ represents the second derivative of $\log P(\theta|X_1)$. Based on previous research (Martens, 2014; Huszár, 2017), $H(\theta^*)$ can be estimated as:

$$\frac{H(\theta^*) - N_k F(\theta^*)}{\lambda_k^p} \approx \sigma_k^p \tag{6}$$

Here, N denotes the number of samples in the current dataset $X_1$, $F(\theta^*)$ represents the empirical Fisher information matrix, and $\lambda_k^p$ is the coefficient used for optimizing the prior distribution. This indicates that there is a certain proportional relationship between the search for the optimal parameter $\theta^*$ and the variance of the Gaussian mixture distribution before optimization. Learning through neural networks to adjust the original spatial distribution parameters can facilitate the search for optimal parameters, prompting us to perform preliminary feature contraction on the original spatial distribution. Furthermore, assuming that the class corresponding to each pixel position $(P_x, P_y)$ in the input image is denoted as $C_k$, the prior probability $P(X_2|\theta)$ can be determined as follows:

$$P(X_2|\theta) = \prod_{k=1}^{K} P(C_k|\theta, P_x, P_y) \tag{7}$$

This suggests that to maximize $P(X_2|\theta)$, the class features associated with each pixel region should demonstrate substantial differentiation and minimal positional coupling. Based on the analysis of equations Equation (6) and Equation (7), compression and sparsity for feature space distribution among different classes can maximize the probability distribution $\log P(X_2|\theta)$ and $\log P(\theta|X_1)$ in incremental segmentation, thereby approaching the performance of joint training.

## 4 FEASIBLE IMPLEMENTATION OF COMPRESSION-SPARSITY PRINCIPLE

### 4.1 BRIEF DESCRIPTION OF THE OVERALL IMPLEMENTATION

Based on the above mathematical analysis, as illustrated in Figure 3, we propose the designs to validate the reliability of the compression-sparsity principle and develop a practical technical solution: 1) Compression: Knowledge gained in each new step, including both category-general and category-specific knowledge, undergoes dimensionality reduction and feature contraction. This compression process concentrates the response regions of features, promoting the generation of compact feature spaces and distinctive feature representations to retain knowledge. 2) Knowledge distillation: By utilizing knowledge transfer, we obtain the feature response distributions from previous steps for the old categories, effectively preventing catastrophic forgetting. 3) Sparsity: The peak values of Gaussian mixture distributions for each category are calculated, and maximum distance constraints

are applied to these peaks. These constraints help allocate spatial distributions with low coupling, thus reducing category confusion. Detailed explanations are provided in the following section.

## 4.2 Implementation details

Considering the necessity of dynamically adjusting the feature distribution, this research aims to continually reconstruct the feature representation to adapt both new and old knowledge. In each new learning step, the high-dimensional knowledge is transformed into a three-dimensional Gaussian mixture distribution (GMD), where the three dimensions correspond to the horizontal pixel position, vertical position, and pixel feature response information in the images. After calculating convex points in feature space, the Euclidean distance between the farthest peak points $P_1$ and $P_2$ in the GMD (GMD) corresponding for class $C_i$ is obtained. Therefore, the relationship between the initial feature $F_t^o$ and the reconstructed feature $F_t^r$ is expressed as follows:

$$F_t^r = \gamma F_t^o + \tau \tag{8}$$

$$\text{subject to} \quad \text{Diam}(F_t^r) < \min \text{D}(P_1^{C_i}, P_2^{C_i}), \quad \forall P_1^{C_i}, P_2^{C_i} \in F_t^o, 0 < i \leq N$$

$$\text{D}(P_1^{C_m}, P_2^{C_n}) > \max \text{D}(P_1^{C_i}, P_2^{C_i}), \quad \forall P_1^{C_m}, P_2^{C_n} \in F_t^r, m \neq n$$

where $\gamma$ and $\tau$ are learnable parameters that satisfy the constraint conditions. Diam represents the diameter of the feature representation. At each learning step, these constraints are designed to facilitate shrinking the reconstructed feature representations by compressing each feature subspace to a diameter smaller than that of all initial feature spaces. Additionally, they ensure the peak distances between different feature spaces exceed the maximum diameter of all initial feature subspaces, hence minimizing coupling. To preserve valuable components of prior knowledge distribution, it is crucial to integrate the compressed and sparse feature distribution $F_t^r$ with the original feature distribution $F_t^o$. This study explores both attention mechanisms and weighted approaches, with the latter being chosen based on comprehensive experimental results to obtain the feature $F_t$ for the current step.

$$F_t = \alpha F_t^o + \beta F_t^r \tag{9}$$

$$P_t = argmax F_t(X_t) \tag{10}$$

$$S_t = [1 + \exp F_t(X_t)]^{-1} \tag{11}$$

where $F_t$, $P_t$, and $S_t$ denote the feature representation, prediction results, and confidence scores produced by the network after learning the $X_t$ data in the t-th step, respectively. Moreover, knowledge transfer is employed to acquire previously learned knowledge of the old categories, referred to as:

$$\widetilde{P}_t = \begin{cases} P_t & \text{when } C = C_t \\ P_{t-1} & \text{when } C = C_u \text{ and } S_{t-1} > 0.7 \end{cases} \tag{12}$$

where $C_t$ and $C_u$ represent the current new class and the regions considered as the background class in the current step, respectively. Subsequently, $\widetilde{P}_t$ and $P_{t-1}$ are optimized based on the following loss function:

$$\mathcal{L}_{CS} = -\frac{1}{||C||} \sum_{i=1}^{||C||} \log \frac{\exp \frac{\widetilde{P_t^i}}{||\widetilde{P_t^i}||} \frac{P_{t-1}^i}{||P_{t-1}^i||}}{\sum_{j=1, j \neq i}^{2||C||} \exp \frac{\widetilde{P_t^i}}{||\widetilde{P_t^i}||} \frac{P_{t-1}^j}{||P_{t-1}^j||}} - \frac{1}{||C||} \log \sum_{i=1}^{||C||} \frac{\exp \widetilde{P_t^i} \otimes Mask_u}{\exp \widetilde{P_t^i}} \tag{13}$$

$$\mathcal{L} = \mathcal{L}_{CS} + \mathcal{L}_{BCE} \tag{14}$$

where $Mask_u$ (Cheng et al., 2021; Zhang et al., 2022b) represents the binary mask of potential target regions in the $X_t$. Binary Cross-entropy (BCE) is a widely used supervised segmentation loss in prior studies (Zhang et al., 2022b; Zhao et al., 2023).

## 5 Experiments

### 5.1 Implementation details.

Following the architectural design of prior works (Cha et al., 2021; Michieli & Zanuttigh, 2021; Cermelli et al., 2020a; Zhang et al., 2023), we incorporate DeepLabV3 and Swin Transformer are used

Table 1: Comparative experiments on VOC dataset (Everingham et al., 2010). Our method achieves significant improvements in plasticity while maintaining stability across diverse configurations.

| | | Backbone | 10-1 (11 steps) | | | 2-2 (10 steps) | | | 15-1 (6 steps) | | | 19-1 (2 steps) | | | 15-5 (2 steps) | | |
|---|---|---|---|---|---|---|---|---|---|---|---|---|---|---|---|---|---|
| | | | 0-10 | 11-20 | All | 0-2 | 3-20 | All | 0-15 | 16-20 | All | 0-19 | 20 | All | 0-15 | 16-20 | All |
| Joint_R | - | Resnet101 | 82.1 | 79.6 | 80.9 | 76.5 | 81.6 | 80.9 | 82.7 | 75.0 | 80.9 | 81.0 | 79.1 | 80.9 | 82.7 | 75.0 | 80.9 |
| Joint_S | - | Swin-B | 82.4 | 83.0 | 82.7 | 75.8 | 83.9 | 82.7 | 83.8 | 79.3 | 82.7 | 82.6 | 84.4 | 82.7 | 83.8 | 79.3 | 82.7 |
| MIB (Cermelli et al., 2020b) | CVPR | Resnet101 | 12.3 | 13.1 | 12.7 | 41.1 | 23.4 | 25.9 | 34.2 | 13.5 | 29.3 | 71.4 | 23.6 | 69.1 | 76.4 | 50.0 | 70.1 |
| SDR (Michieli & Zanuttigh, 2021) | CVPR | Resnet101 | 32.1 | 17.0 | 24.9 | 13.0 | 5.1 | 6.2 | 44.7 | 21.8 | 39.2 | 69.1 | 32.6 | 67.4 | 75.4 | 52.6 | 70.0 |
| PLOP (Douillard et al., 2021) | CVPR | Resnet101 | 44.0 | 15.5 | 30.4 | 24.1 | 11.9 | 13.6 | 65.1 | 21.1 | 54.6 | 75.4 | 37.4 | 73.6 | 75.7 | 51.7 | 70.0 |
| REMINDER (Phan et al., 2022) | CVPR | Resnet101 | - | - | - | - | - | - | 68.3 | 27.7 | 58.6 | 76.5 | 32.3 | 74.4 | 76.1 | 50.7 | 70.1 |
| RCIL (Zhang et al., 2022a) | CVPR | Resnet101 | 55.4 | 15.1 | 36.2 | 28.3 | 19.0 | 20.3 | 70.6 | 23.7 | 59.4 | 68.5 | 12.1 | 65.8 | 78.8 | 52.0 | 72.4 |
| SSUL (Cha et al., 2021) | NIPS | Resnet101 | 74.0 | 53.2 | 64.1 | - | - | - | 78.4 | 49.0 | 71.4 | 77.8 | 49.8 | 76.5 | 78.4 | 55.8 | 73.0 |
| MicroSeg (Zhang et al., 2022b) | NIPS | Resnet101 | 77.2 | 57.2 | 67.7 | 60.0 | 50.9 | 52.2 | 81.3 | 52.5 | 74.4 | 79.3 | 62.9 | 78.5 | 82.0 | 59.2 | 76.6 |
| SSUL+ (Cha et al., 2021) | NIPS | Swin-B | 74.3 | 51.0 | 63.2 | 60.3 | 40.6 | 43.4 | 78.1 | 33.4 | 67.5 | 80.8 | 31.5 | 78.5 | 79.7 | 55.3 | 73.9 |
| MicroSeg+ (Zhang et al., 2022b) | NIPS | Swin-B | 73.5 | 53.0 | 63.7 | 64.8 | 43.4 | 46.5 | 80.5 | 40.8 | 71.0 | 79.0 | 25.3 | 76.4 | 81.9 | 54.0 | 75.3 |
| EWF (Xiao et al., 2023) | CVPR | Resnet101 | 71.5 | 30.3 | 51.9 | - | - | - | 77.7 | 32.7 | 67.0 | 77.9 | 6.7 | 74.5 | - | - | - |
| LGKD (Yang et al., 2023) | ICCV | Resnet101 | - | - | - | - | - | - | 70.6 | 30.9 | 61.1 | 77.3 | 42.9 | 75.7 | 79.5 | 54.8 | 73.6 |
| IDEC (Zhao et al., 2023) | TPAMI | ResNet101 | 70.7 | 46.3 | 59.1 | - | - | - | 77.0 | 36.5 | 67.4 | - | - | - | 78.0 | 51.8 | 71.8 |
| GSC (Cong et al., 2023) | TMM | ResNet101 | 50.6 | 17.3 | 34.7 | - | - | - | 72.1 | 24.4 | 60.7 | 76.9 | 42.7 | 75.3 | 78.3 | 54.2 | 72.6 |
| CoMFormer (Cermelli et al., 2023) | CVPR | ResNet101 | - | - | - | - | - | - | 49.0 | 23.3 | 42.9 | 75.4 | 24.1 | 72.9 | 74.7 | 48.5 | 68.4 |
| CoinSeg (Zhang et al., 2023) | ICCV | Swin-B | 80.1 | 60.0 | 70.5 | 70.1 | 63.3 | 64.3 | 82.7 | 52.5 | 75.5 | 81.5 | 44.8 | 79.8 | 82.1 | 63.2 | 77.6 |
| CoMasTRe (Gong et al., 2024) | CVPR | ResNet101 | - | - | - | - | - | - | 69.8 | 43.6 | 63.5 | 75.1 | 69.5 | 74.9 | 79.7 | 51.9 | 73.1 |
| Ours | - | ResNet101 | 74.1 | 57.7 | 66.3 | 56.4 | 55.1 | 55.3 | 77.7 | 52.2 | 71.6 | 76.6 | 61.4 | 75.9 | 78.3 | 55.5 | 72.9 |
| Ours ($\alpha$=0.8,$\beta$=0.2) | - | Swin-B | 80.3 | 69.8 | 75.3 | 68.0 | 69.5 | 69.3 | 83.6 | 64.3 | 79.0 | 82.4 | 67.8 | 81.7 | 78.6 | 70.3 | 76.6 |
| Ours ($\alpha$=0.2,$\beta$=0.8) | - | Swin-B | 81.7 | 71.7 | 76.9 | 66.7 | 69.1 | 68.8 | 83.4 | 66.7 | 79.4 | 82.0 | 75.5 | 81.7 | 83.7 | 71.5 | 80.8 |

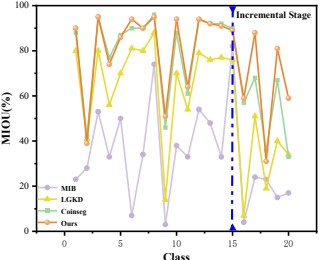

Figure 4: Line chart comparing MIoU performance across all classes in the 15-1 incremental setting. Our method demonstrates a significant improvement in MIoU values across multiple classes, particularly evident during the incremental stage.

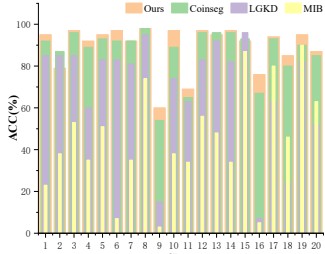

Figure 5: Bar chart comparing accuracy performance across all classes in the 15-1 incremental setting. Our method (shown in orange) attains superior accuracy across most classes, notably excelling in the five latest learning classes (16-20).

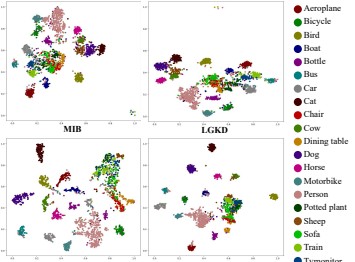

Figure 6: Visualization of feature distribution using T-SNE in the 15-1 incremental setting. Our approach shows more noticeable intra-class clustering and inter-class dispersion. In the incremental stage, ours exhibits reduced confusion among classes.

as components the architecture (Chen et al., 2016; Liu et al., 2021) . The ADAMW optimizer is employed for training optimization (Loshchilov & Hutter, 2017), applying different learning rates for various modules. To ensure a fair comparison, we adopt the same memory sampling strategy (Cha et al., 2021)during training. Additionally, we include the widely used ResNet architecture (Szegedy et al., 2016) to evaluate its performance in joint training and under various incremental configurations. More details and code are provided in the supplementary materials.

## 5.2 COMPARATIVE EXPERIMENT

**Quantitative Evaluation.** Table 1 presents the performance comparison based on MIoU for various methods and incremental settings. As the number of steps increases, the challenge of achieving plasticity performance becomes greater. Due to the design principles based on compression-sparsity techniques discussed in this paper, we observe a significant improvement in plasticity, especially with hyper-parameters $\alpha$=0.2/$\beta$=0.8 in the challenging 10-1 setting, where the plasticity rises by 11.7% compared to previous methods. Besides, enhancements of 9.8%, 6.2%, 11.8%, and 7.1% in plasticity are obtained with hyper-parameters $\alpha$=0.8/$\beta$=0.2 in the 10-1, 2-2, 15-1, and 15-5 incremental configurations, respectively. Table 2 illustrates the learning performance of the challenging incremental configuration of another dataset, which spans a total of 11 steps. In the incremental configuration of 100-5 within ADE20K dataset, our method shows a notable degree of performance improvement. From Figure 4, it is evident that even after five steps without transferring all data

Table 2: Comparison of our method with recent approaches, on the challenging 100-5 setting of ADE dataset (Zhou et al., 2017)) in terms of MIoU. Our method demonstrates consistent performance improvements in both the base stage and the incremental stage.

| | Joint_S | SDR | PLOP | REMINDER | RCIL | SSUL | MicroSeg | SSUL+ | Microseg+ | EWF | IDEC | CoMFormer | CoMadTRe | Ours |
|---|---|---|---|---|---|---|---|---|---|---|---|---|---|---|
| 0-100 | 43.5 | 36.0 | 39.1 | 36.1 | 38.5 | 39.9 | 40.4 | 41.3 | 41.2 | 41.4 | 39.2 | 34.4 | 40.8 | **41.6** |
| 101-150 | 30.6 | 5.7 | 7.8 | 16.4 | 11.5 | 17.4 | 20.5 | 16.0 | 21.0 | 13.4 | 14.6 | 15.9 | 15.8 | **25.5** |
| All | 39.2 | 26.0 | 28.7 | 29.5 | 29.6 | 32.4 | 33.8 | 32.9 | 34.5 | 32.1 | 31.0 | 28.3 | 32.5 | **36.3** |

Figure 7: Visual comparison on 15-1 setting. Our method exhibits less knowledge confusion in the base stage and demonstrates stronger capabilities for new classes in the incremental stages.

from the first fifteen classes, our method maintains remarkable segmentation performance. Additionally, the results from classes 16 to 20 show that our method exhibits superior adaptability for learning new classes. Figure 5 further illustrates that our method shows a significant improvement in accuracy performance for each class, highlighting the effectiveness of dynamic learning manner based on the compression-sparsity principle.

**Qualitative Evaluation.** From Figure 6, it demonstrates that our method exhibits a more concentrated intra-class and a sparser inter-class distribution in the base stage (first fifteen classes). This illustrates that the proposed method, based on the compression-sparsity principle, can effectively modify the distribution area and spacing of features. Moreover, during the incremental stage, where one new class is learned at a time, the overlap among newly added classes is minimal. Although the inherent incompleteness of the data results in the inter-class distances is not strongly sparse across different stages, this low coupling still allows for good learning performance for new classes. To depicts the visual comparison, as shown in Figure 7, we employ publicly available codes and training strategies from MIB (Cermelli et al., 2020b) and LGKD (Yang et al., 2023) to evaluate the segmentation results for the 15-1 configuration. Furthermore, we retrain the Coinseg (Zhang et al., 2023) method using the same backbone and memory sampling strategy (Cha et al., 2021) to compare its visual results. In both the base stage for old classes and the incremental for new classes, our method demonstrates superior pixel-level segmentation accuracy and category correctness.

### 5.3 ABLATION EXPERIMENT AND DISCUSSION

**Effectiveness of compressioin-sparsity based algorithm.** In Table 3, we show the ablation experiments conducted on the VOC dataset for the incremental settings 19-1 and 10-1. By observing the results of groups 1, and 5, it is evident that compression and sparsity significantly contribute to balancing stability and plasticity. Based on the performance of groups 5 and 8, whether integrating knowledge distillation (KD) or not, compression and sparsity have the capacity to balance stability and plasticity. We maintain the KD module to ensure superior stability. Observations from the results from groups 7 and 8, the degradation in performance is more obvious in 19-1 compared to 10-1

Table 3: Abaltion studies of compression-sparsity based algorithm. Compression (C) and sparsity (S) play a crucial role in learning knowledge.

| | KD | C | S | 0-19 | 20 | All | 0-10 | 11-20 | All |
|---|---|---|---|---|---|---|---|---|---|
| 01 | × | × | × | 73.0 | 37.8 | 71.3 | 7.2 | 13.0 | 10.0 |
| 02 | ✓ | × | × | 81.9 | 36.2 | 79.7 | 76.6 | 57.3 | 67.4 |
| 03 | × | ✓ | × | 82.0 | 41.3 | 80.1 | 77.4 | 57.1 | 67.7 |
| 04 | × | × | ✓ | 81.9 | 40.8 | 79.9 | 71.4 | 55.8 | 64.0 |
| 05 | × | ✓ | ✓ | 82.2 | 70.5 | 81.6 | 79.9 | 70.7 | 75.5 |
| 06 | ✓ | × | ✓ | 82.0 | 60.7 | 81.0 | 81.2 | 67.8 | 74.8 |
| 07 | ✓ | ✓ | × | 81.8 | 65.5 | 81.0 | 80.6 | 70.6 | 75.8 |
| 08 | ✓ | ✓ | ✓ | 82.4 | 67.8 | 81.7 | 80.3 | 69.8 | 75.3 |

Table 4: Comparison of feature space fusion methods. The weighted approach exhibits superior overall performance.

| | | 10-1 | 2-2 | 15-1 | 19-1 | 15-5 |
|---|---|---|---|---|---|---|
| Attention Mechanism | Base Stage | 80.1 | 48.8 | 80.6 | 80.7 | 70.7 |
| | Incremental Stage | 68.8 | 67.8 | 61.9 | 61.3 | 69.3 |
| | All | 74.4 | 65.9 | 75.9 | 79.8 | 70.3 |
| Weighted Approach | Base Stage | 80.3 | 68.0 | 83.6 | 82.4 | 78.6 |
| | Incremental Stage | 69.8 | 69.5 | 64.3 | 67.8 | 70.3 |
| | All | 75.3 | 69.3 | 79.0 | 81.7 | 76.6 |

Table 5: Impact of $\alpha$ and $\beta$ parameters in Equation (9). $\alpha$ and $\beta$ can effectively balance the stability of the base stage and the plasticity of the incremental stage across diverse parameter configurations.

| | | $\alpha = 0.2, \beta = 0.8$ | | | $\alpha = 0.5, \beta = 0.5$ | | | $\alpha = 0.8, \beta = 0.2$ | | |
|---|---|---|---|---|---|---|---|---|---|---|
| | Steps | Base Stage | Incremental Stage | All | Base Stage | Incremental Stage | All | Base Stage | Incremental Stage | All |
| 10-1 | 11 | 81.7 | 71.7 | 76.9 | 81.5 | 72.5 | 77.2 | 80.3 | 69.8 | 75.3 |
| 2-2 | 10 | 66.7 | 69.1 | 68.8 | 69.5 | 69.9 | 69.8 | 68.0 | 69.5 | 69.3 |
| 15-1 | 6 | 83.4 | 66.7 | 79.4 | 81.7 | 65.0 | 77.7 | 83.6 | 54.3 | 76.6 |
| 19-1 | 2 | 82.0 | 75.5 | 81.7 | 82.2 | 61.0 | 81.2 | 82.4 | 67.8 | 81.7 |
| 15-5 | 2 | 83.7 | 71.5 | 80.8 | 83.0 | 70.8 | 80.1 | 78.6 | 70.3 | 76.6 |

incremental operations in the absence of sparsity. This disparity arises because the compressed operations in 10-1 learning undergo efficient iterative compression with more steps, thereby facilitating plasticity. Considering the performance of the base and incremental stage on multiple incremental configurations, the combined use of knowledge distillation, compression, and sparsity can be more conducive to balancing stability and plasticity.

**Integration Approach: Attention mechanism VS weighted approach.** To balance the distribution of feature space between old knowledge and new knowledge, we explore two commonly used feature fusion approaches in this paper: the attention-based method (Vaswani et al., 2017) and the weighted-based method (Lee et al., 2017). Across five different incremental settings, the weighted approach consistently demonstrates superior overall performance, as shown in Table 4.Therefore, in Equation (9), we employ the weighted approach to improve performance in alignment that aligns with the principles of compression and sparsity.

**Effectiveness of weighted coefficient.** To assign appropriate values in Equation (9), we conduct three sets of experiments, as shown in Table 5. A higher $\alpha$ value indicates an increased presence of original features in the fusion feature, while a higher $\beta$ value signifies a greater proportion of reconstructed features. Specifically, when setting $\alpha$ to 0.2 and $\beta$ to 0.8, our method demonstrates a notable performance advantage on both old and new categories. Through our experiments, we observe that for datasets with a larger number of categories like ADE20K, preserving more original features in the fusion feature is advantageous for incremental segmentation. Though models with hyper-parameters $\alpha$=0.2/$\beta$=0.8 achieve best results in the VOC dataset, we would like to show the robustness of our method on variate datasets for fair comparisons in a consistent manner. Thus, $\alpha$ and $\beta$ are set to 0.8 and 0.2 in this paper for qualitative and quantitative analysis.

# 6 CONCLUTION

In this paper, we conduct a mathematical analysis focusing on the good stability but limited plasticity in the current incremental segmentation learning. We find that dynamically adjusting the distribution of new and old knowledge based on the compression-sparsity principle can promote the balance between stability and plasticity. Building upon the investigation of Gaussian mixture distribution, we propose a viable algorithm. In contrast to existing incremental segmentation learning methods, we advocate for the adaptation of prior knowledge to newly acquired knowledge, rather than retaining parameters statically or preserving the invariance of the old space. This adaptive transformation enhances feature compression and promotes sparse space distribution, facilitating the extraction of discriminative features while maintaining stability in prior stages and improving adaptability to new stages. Through comparative experiments and ablation experiments conducted across five different difficulty levels in the incremental learning setups, we comprehensively demonstrate the feasibility of the compression-sparsity principle.

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

APPENDIX

## A    COMPRESSION-SPARSITY BASED ALGORITHM

---
**Algorithm 1** Feasible implementation pseudocode for compression-sparsity principles.

---
**Input:** $P$: Three-dimensional point set
**Output:** $\widetilde{P}_t$: Optimized multi-class segmentation prediction
1: *Initialize an empty list $P_{peaks}$.*
2: **RecursivePeakFinder**($P, P_{\text{peaks}}$)
3:      **IF** $|P| = 1$ **THEN**
4:          **Add point $P[0]$ to the list** $P_{\text{peaks}}$
5:      **ELSE**
6:          **Compute the midpoint index** $mid$ **of** $P$
7:          **IF** $P[mid]$ **is a peak THEN**
8:              **Add point** $P[mid]$ **to the list** $P_{\text{peaks}}$
9:          **IF** $P[mid-1] > P[mid]$ **THEN**
10:              **RecursivePeakFinder**($P[0 : mid - 1], P_{\text{peaks}}$)
11:          **IF** $P[mid+1] > P[mid]$ **THEN**
12:              **RecursivePeakFinder**($P[mid + 1 : |P| - 1], P_{\text{peaks}}$)
13: **Compression and Sparsity** with  Equation (8)
14:  **Fusion of reconstructed and original feature distribution** with  Equation (9)
15: **Knowledge Distillation** with  Equation (10) -  Equation (12)
16: **Optimize the parameters** with  Equations (13) and (14)

---

Algorithm 1 provides a logical demonstration of the pseudo-code for the incremental segmentation architecture implemented based on the compression-sparsity principle. To establish initial compression and coefficient soft constraints during incremental segmentation, we first employ RecursivePeakFinder to identify peaks within each Gaussian distribution. Subsequently, utilizing  Equation (8), we preliminarily compress and sparsify the feature representation, significantly facilitating the plasticity of new knowledge. To balance the initial knowledge and reconstructed knowledge, we integrate the reconstructed features with the original ones according to  Equation (9). To prevent catastrophic forgetting, we transfer high-confidence knowledge from previous categories to the current stage, ensuring that the high confidence of old categories can still be maximally retained. Finally, we optimize predictions by considering both old and new knowledge using  Equations (13) and (14).

## B    MORE IMPLEMENTATION DETAILS

### B.1    EXPERIMENT DATASET

This paper utilizes the VOC 2012 and ADE20K. Apart from the background category, the VOC dataset consists of a total of twenty categories, namely Aeroplane, Bicycle, Bird, Boat, Bottle, Bus, Car, Cat, Chair, Cow, Dining table, Dog, Horse, Motorbike, Person, Potted plant, Sheep, Sofa, Train, and TV monitor. The division of training, validation, and test sets in the dataset follows the original segmentation settings. The original VOC 2012 dataset comprised 1464 training samples, 1449 validation samples, and 1456 test samples. The augmented dataset includes 10582 training samples, 1449 validation samples, and 1456 test samples. The results in the paper are based on the latter for training. The ADE dataset features 150 categories for incremental segmentation, sourced from the SUN dataset (2010, Princeton University) and the Places dataset (2014, MIT). Currently, there is no public test set available for this dataset. As a result, the validation set of the original dataset is repurposed as the test set, comprising a total of 2000 images. The training set contains a total of 20,210 images. The images in the datasets have been subjected to anonymization procedures, such as facial and license plate blurring, along with the elimination of private information.

B.2    INCREMENTAL SETTING

Building upon previous work, we explore five distinct incremental configurations for the VOC dataset, including 10-1, 2-2, 15-1, 19-1, and 15-5. For the ADE dataset, we establish two different incremental setups: 100-5 and 100-10. These varied configurations correspond to different numbers of learning categories in the base stage and incremental stage. For instance, in the 10-1 setup, the base stage involves learning 10 categories, with each subsequent incremental stage adding one new category. The data from the previous ten categories cannot all participate in joint training, leading to a total of 11 learning steps. Similarly, in the 2-2 setting, the base stage includes learning two categories, with each subsequent incremental stage also involving two categories, totaling 10 steps. A higher number of steps indicates a more challenging setting for enhancing the plasticity of new classes while maintaining the stability of old classes. In real-world scenarios, data arrives intermittently, similar to incremental learning settings. To address the challenge of learning new incoming data without extensive time and computational resources, while simultaneously preserving the performance of old data, we conduct tests and research on a total of seven different incremental learning configurations for the VOC and ADE datasets.

B.3    TRAINING DETAILS

When training the incremental configurations 10-1, 2-2, 15-1, 19-1, and 15-5, we utilize the training set of the VOC dataset. Notably, each training session loads data corresponding to specific categories based on the incremental setting, rather than the entire training set. To enhance training efficiency, images from the VOC dataset are cropped to a resolution of 513x513 due to high data resolution. We also integrate augmented data following previous works Data preprocessing involves techniques such as resizing, scaling, cropping, flipping, and normalization. Normalization is performed using a mean of [0.485, 0.456, 0.406] and a variance of [0.229, 0.224, 0.225]. During training, learning rates vary across different modules, and we employ the AdamW optimizer utilized. Each incremental configuration undergoes 50 epochs of training on a single 3090 GPU for both the base and incremental stages. For the 100-5 configuration, we use the training set of ADE20K, selectively loading data based on the incremental setting in each training stage. During training, we implement a replay buffer following prior researches (Cha et al., 2021; Zhang et al., 2022b), which restricts the storage of instances per class to a maximum of ten. The data preprocessing, learning rates, and optimizer settings mirrored those described earlier. Each incremental configuration is trained for 100 epochs using two A100 GPUs, and the implementation is carried out with PyTorch.

B.4    TESTING DETAILS

After learning twenty different classes based on various incremental configurations, including 10-1, 2-2, 15-1, 19-1, and 15-5, we load the best pth file generated from the final step to evaluate the MIoU and ACC performance across all classes in the VOC test set. In this study, we measure the catastrophic forgetting resistance (stability) of old classes by evaluating the MIoU and ACC performance on test data corresponding to the classes learned in the base stage. Additionally, we evaluate the learning ability (plasticity) of new classes by testing the MIoU and ACC performance on data involving classes in the incremental stage. Before inference, the test data must undergo normalization to ensure compatibility with the algorithm. Similarly, for the incremental configuration 100-5, we measure the corresponding MIoU and ACC metrics on the ADE20K test set after completing learning all incremental steps. All experiments are conducted using PyTorch on 3090 GPU and A100 GPU.

# C    ADDITIONAL EXPERIMENT RESULTS AND DISCUSSION.

**Benefits of the Compression-Sparsity Principle.** Implementation of the Compression-Sparsity Principle in incremental segmentation effectively addresses the challenge of limited performance in new classes while preserving the performance of old classes. As shown in Table 6, we have compiled Mean Intersection over Union (MIoU) and accuracy (Acc) for 21 subclasses in a 15-1 incremental configuration, comparing three typical methods with our approach. The averages calculated in the table represent the mean MIoU and accuracy for individual categories, facilitating performance comparisons among various methods. It is evident from the table that our approach

Table 6: Comparison with recent approaches based on Mean Intersection over Union (MIoU) and accuracy (ACC) across multiple subclasses. Benefiting from the Compression-Sparsity principle, our method shows significant plasticity performance improvements in the incremental stage (last five categories) while maintaining stability in handling old classes.

| | MIB | | | LGKD | | | Coinseg | | | Ours | | |
|---|---|---|---|---|---|---|---|---|---|---|---|---|
| | MIoU | Acc | Average | MIoU | Acc | Average | MIoU | Acc | Average | MIoU | Acc | Average |
| Background | 85.46 | 90.88 | 88.17 | 89.15 | 94.55 | 91.85 | 90.65 | 93.39 | 92.02 | 91.09 | 93.56 | **92.33** |
| Aeroplane | 23.61 | 23.67 | 23.64 | 80.41 | 85.23 | 82.82 | 88.80 | 92.33 | 90.57 | 90.78 | 95.29 | **93.04** |
| Bicycle | 28.31 | 38.81 | 33.56 | 41.07 | 85.13 | 63.10 | 42.81 | 87.86 | **65.34** | 39.88 | 79.18 | 59.53 |
| Bird | 53.33 | 53.55 | 53.44 | 80.03 | 85.78 | 82.91 | 95.42 | 96.95 | 96.19 | 95.24 | 97.20 | **96.22** |
| Boat | 33.94 | 35.39 | 34.67 | 56.33 | 60.17 | 58.25 | 77.89 | 89.41 | **83.65** | 74.09 | 92.49 | 83.29 |
| Bottle | 50.43 | 51.45 | 50.94 | 70.65 | 83.58 | 77.12 | 87.88 | 93.81 | **90.85** | 86.26 | 95.43 | **90.85** |
| Bus | 7.38 | 7.38 | 7.38 | 81.96 | 83.68 | 82.82 | 90.55 | 92.57 | 91.56 | 94.61 | 97.17 | **95.89** |
| Car | 34.91 | 35.02 | 34.97 | 80.37 | 81.93 | 81.15 | 90.67 | 92.69 | **91.68** | 90.18 | 92.65 | 91.42 |
| Cat | 74.32 | 74.58 | 74.45 | 88.29 | 95.21 | 91.75 | 96.48 | 98.21 | **97.35** | 95.93 | 98.29 | 97.11 |
| Chair | 3.49 | 3.53 | 3.51 | 14.90 | 15.47 | 15.19 | 46.69 | 54.83 | 50.76 | 51.07 | 60.04 | **55.56** |
| Cow | 38.03 | 38.69 | 38.36 | 70.75 | 74.66 | 72.71 | 88.09 | 89.45 | 88.77 | 94.79 | 97.55 | **96.17** |
| Dining table | 33.09 | 34.12 | 33.61 | 54.87 | 63.18 | 59.03 | 61.25 | 65.71 | 63.48 | 64.79 | 69.39 | **67.09** |
| Dog | 54.64 | 56.83 | 55.74 | 79.50 | 83.18 | 81.34 | 94.91 | 96.95 | 95.93 | 94.50 | 97.50 | **96.00** |
| Horse | 48.30 | 48.81 | 48.56 | 76.96 | 92.64 | 84.80 | 92.94 | 96.01 | **94.48** | 92.38 | 95.86 | 94.12 |
| Motorbike | 33.96 | 34.35 | 34.16 | 77.74 | 82.78 | 80.26 | 92.46 | 96.00 | 94.23 | 91.48 | 97.00 | **94.24** |
| Person | 82.67 | 87.95 | 85.31 | 76.90 | 96.32 | 86.61 | 90.53 | 93.00 | **91.77** | 89.74 | 92.58 | 91.16 |
| Potted plant | 4.93 | 5.55 | 5.24 | 7.57 | 7.75 | 7.66 | 57.72 | 67.72 | 62.72 | 59.83 | 76.68 | **68.26** |
| Sheep | 24.71 | 80.74 | 52.73 | 51.28 | 63.89 | 57.59 | 68.47 | 93.46 | 80.97 | 88.52 | 94.71 | **91.62** |
| Sofa | 23.18 | 46.31 | 34.75 | 19.45 | 24.19 | 21.82 | 36.57 | 80.16 | 58.37 | 31.97 | 85.24 | **58.61** |
| Train | 15.40 | 90.38 | 52.89 | 40.73 | 82.54 | 61.64 | 67.19 | 90.85 | 79.02 | 81.13 | 95.42 | **88.28** |
| Tv monitor | 17.95 | 63.62 | 40.79 | 34.42 | 52.22 | 43.32 | 33.61 | 85.52 | 59.57 | 59.96 | 87.44 | **73.70** |

Table 7: Comparison of our method with recent approaches, on the challenging 100-10 setting of ADE dataset (Zhou et al., 2017)) in terms of MIoU. In the incremental stage, our method demonstrates a certain degree of performance improvement.

| | Joint | SDR | MIB | PLOP | Reminder | RCIL | SSUL | Microseg | SSUL+ | MicroSeg+ | EWF | LGKD | IDEC | GSC | CoMFormer | CoMasTRe | Ours |
|---|---|---|---|---|---|---|---|---|---|---|---|---|---|---|---|---|---|
| 0-100 | 43.5 | 28.9 | 38.2 | 40.5 | 39.0 | 39.3 | 40.2 | 41.5 | 40.7 | 41.0 | 41.5 | 42.0 | 42.3 | 40.8 | 36.0 | **42.8** | 41.6 |
| 101-150 | 30.6 | 7.4 | 11.1 | 13.6 | 21.3 | 17.7 | 18.8 | 21.6 | 19.0 | 22.6 | 16.3 | 20.4 | 17.6 | 17.6 | 17.1 | 15.8 | **25.5** |
| All | 39.2 | 21.8 | 29.2 | 31.6 | 33.1 | 32.2 | 33.1 | 34.9 | 33.5 | 34.9 | 33.2 | 34.9 | 34.1 | 33.1 | 29.7 | 33.9 | **36.3** |

demonstrates superior performance across multiple categories among the first sixteen. Particularly, our method shows significant performance improvements in the categories learned during the final five incremental stages, specifically in the Potted plant, Sheep, Sofa, Train, and TV monitor categories, with increases of 5.54%, 10.65%, 0.24%, 9.26%, and 14.13%, respectively. Table 7 illustrates the performance comparison under the 100-10 incremental setting on the ADE20K dataset. Compared to the suboptimal method, we achieve a performance improvement of 2.9% on new categories (101-150). These notable enhancements in adaptability can be attributed to our method's capability to provide more discriminative features, which aids in reducing confusion among category features and shapes a more segmentation-friendly feature space.

As shown in Figure 8, we visualize feature attention maps with (columns four and five) and without the Compression-Sparsity method (columns two and three). It is worth noting that this visualization does not represent features from the final layer of the network, but rather from a feature layer selected for compression and sparsity operations. Columns two and four illustrate the effects after averaging multiple channels, while columns three and five display the results after summing features from multiple channels and overlaying them on the original image. It is evident that the high-heat response regions of features become more enriched in both quantity and area on most images following the incorporation of compression and sparsity. This observation further validates that the Compression-Sparsity method can provide more discriminative features, thereby promoting a balance between stability and plasticity.

Additionally, Figure 9 illustrates the test results of our method compared to recent methods across all test sets in the 15-1 incremental configuration. The results indicate that our method exhibits fewer category confusions after learning new classes. Furthermore, our approach demonstrates enhanced adaptability towards new categories. Table 8 presents a statistical analysis of the Mean Intersection over Union (MIoU) values across multiple incremental steps under a 10-1 incremental setting. The two compared groups are G4 (an incremental algorithm without the Compression-Sparsity operation in ablation experiments) and G7 (an algorithm incorporating the Compression-Sparsity operation).

Table 8: Comparison between the method with compression-sparsity (G7) and the method without compression-sparsity (G4). By analyzing the MIoU values of multiple steps in the intricate 10-1 incremental setting, the incorporation of the compression-sparsity principle facilitates the assimilation of knowledge for new categories in the incremental stage.

| Step | 1 | 2 | 3 | 4 | 5 | 6 | 7 | 8 | 9 | 10 |
|------|------|------|------|------|------|------|------|------|------|------|
| G4 | 22.7 | 63.2 | 67.9 | 71.9 | 76.0 | 70.1 | 66.0 | 60.3 | 61.4 | 57.3 |
| G7 | 44.6 | 72.6 | 76.8 | 79.6 | 77.6 | 74.1 | 73.7 | 65.7 | 70.6 | 69.8 |
| | ↑21.9 | ↑9.4 | ↑8.9 | ↑7.7 | ↑1.6 | ↑4 | ↑7.7 | ↑5.4 | ↑9.2 | ↑12.5 |

Table 9: Comparative experiments conducted without replay in a 2-2 incremental setup. It is demonstrated that even without replay, the compression-sparsity approach exhibits strong learning capabilities for new classes (3-20).

| | Joint | MIB | SDR | PLOP | RCIL | SSUL+ | Microseg+ | Coinseg | Ours |
|------|-------|------|------|------|------|-------|-----------|---------|------|
| 0-2 | 75.8 | 41.1 | 13.0 | 24.1 | 28.3 | 60.3 | 64.8 | 70.1 | **70.6** |
| 3-20 | 83.9 | 23.4 | 5.1 | 11.9 | 19.0 | 40.6 | 43.4 | 63.3 | **65.8** |
| All | 82.7 | 25.9 | 6.2 | 13.6 | 20.3 | 43.4 | 46.5 | 64.3 | **66.5** |

A direct comparison reveals that the incorporation of the Compression-Sparsity operation effectively enhances the plasticity of new categories to a greater extent. Specifically, in the first step, performance improvement is notably increased by $21.9\%$ compared to the absence of the Compression-Sparsity method.

**Methods without Replay.** To validate the effectiveness of our method in a no-replay scenario, we conduct experiments under a zero-replay setting and compare our approach to recent state-of-the-art methods in a 2-2 incremental configuration. As illustrated in Table 9, our method demonstrates a balanced performance in terms of stability and plasticity, even in the absence of replay. Particularly, when compared to the previously second-best method, our approach demonstrates an improvement of $2.2\%$ in overall category performance. While our method achieves significant performance without replay in incremental segmentation, we still recommend utilizing a small amount of replay, where hardware allows, to further enhance performance.

# D  LIMITATION

Although this study demonstrates a significant enhancement in the plasticity of new classes through incremental learning utilizing the compression-sparsity principle, the spatial separation between the class centers learned during the incremental step remains relatively close, as indicated by the t-SNE plot. While this distance is sufficient to support notable enhancements in MIoU and ACC performance, it also indicates the need for further efforts to increase the distribution gap between new classes in future work. To maintain a balance between stability and plasticity, classes within the same step undergo more substantial adaptive changes, resulting in relatively smaller fluctuations in adaptability among classes across different steps. This limitation primarily arises because the data involved in loss calculations mainly consists of data from the new classes in the current step, where the influence of past data knowledge during the incremental stages mainly focuses on knowledge distillation rather than spatial sparsity. Therefore, we will reassess how classes in different steps can achieve greater sparsity in the distribution with minimal replay during the adaptive change process in future research.

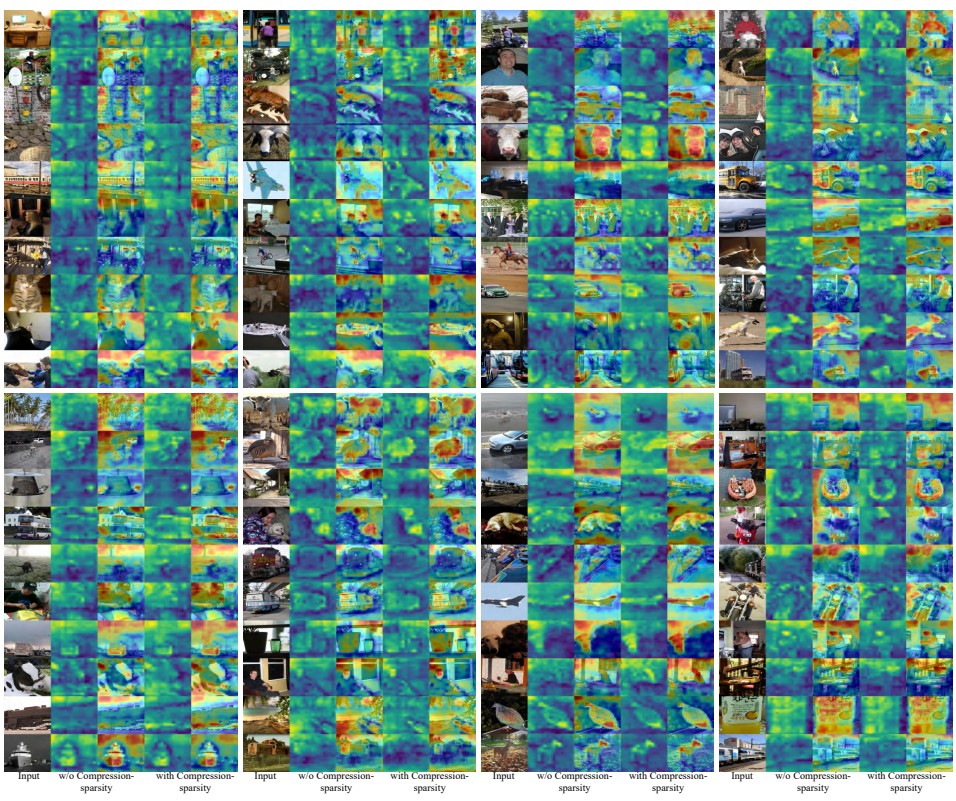

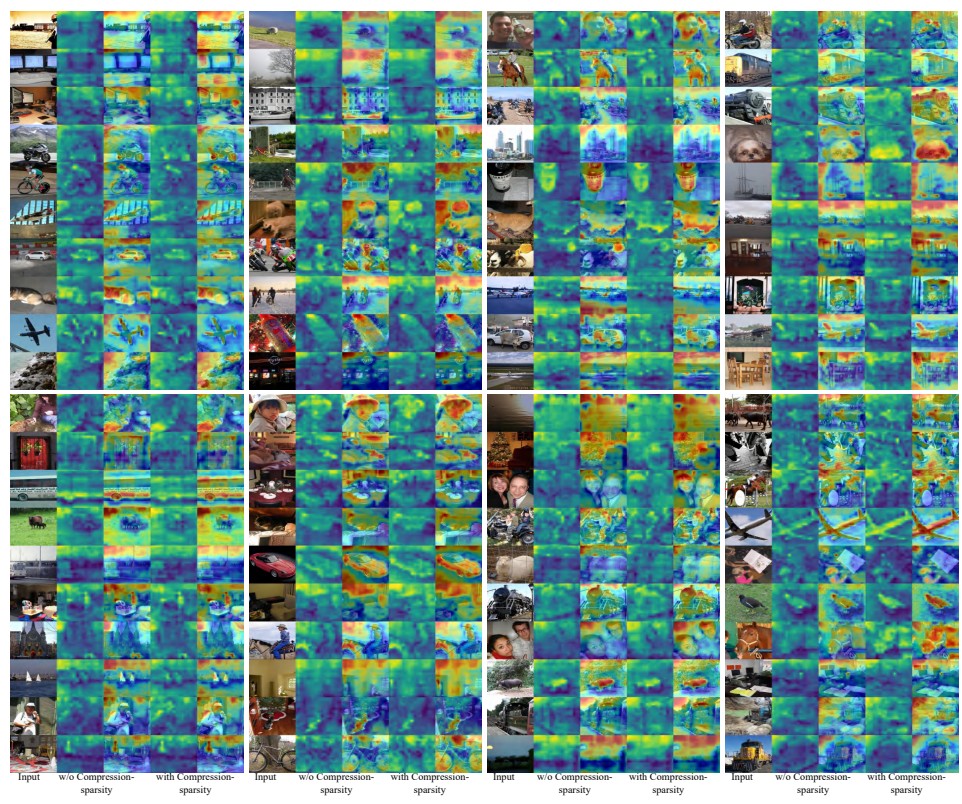

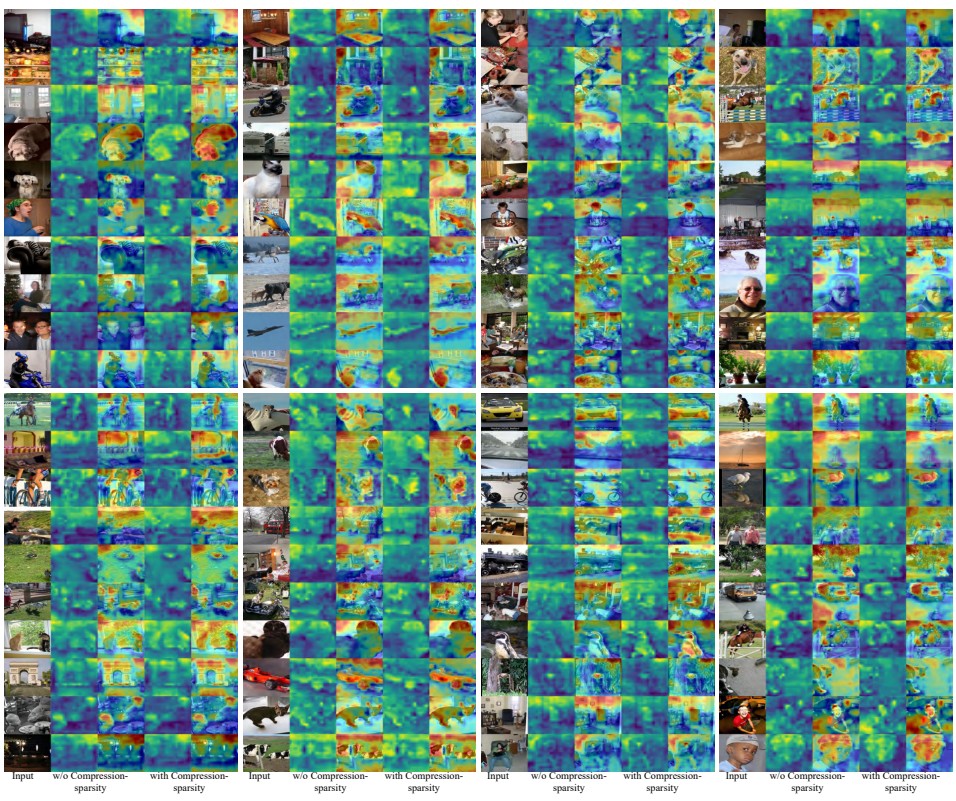

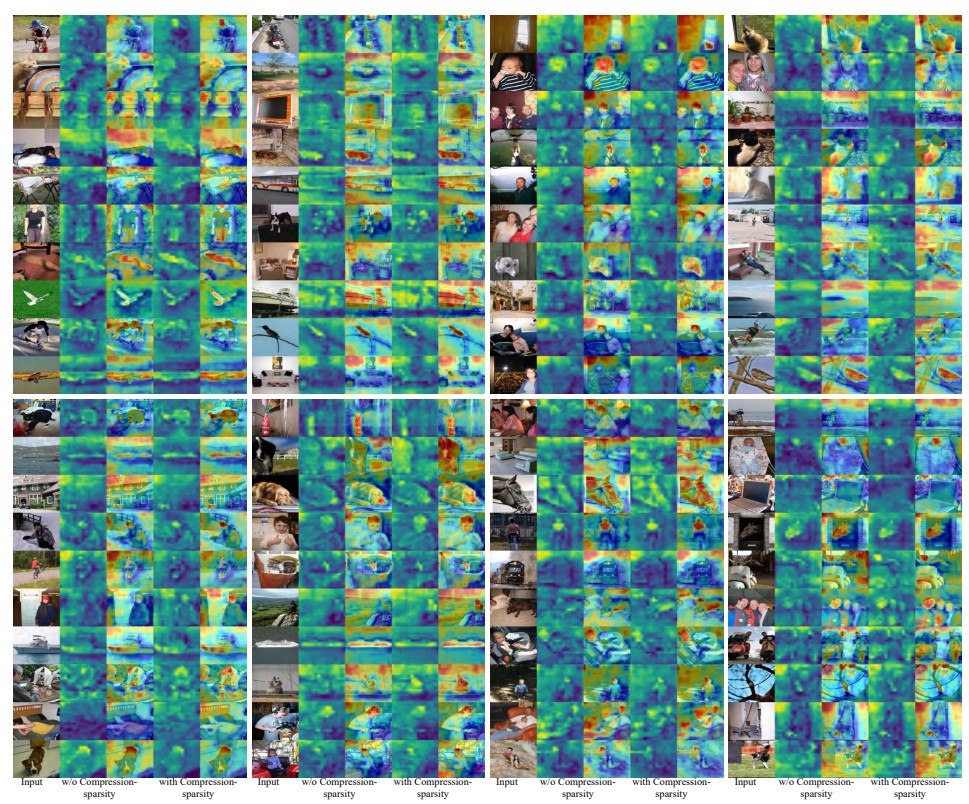

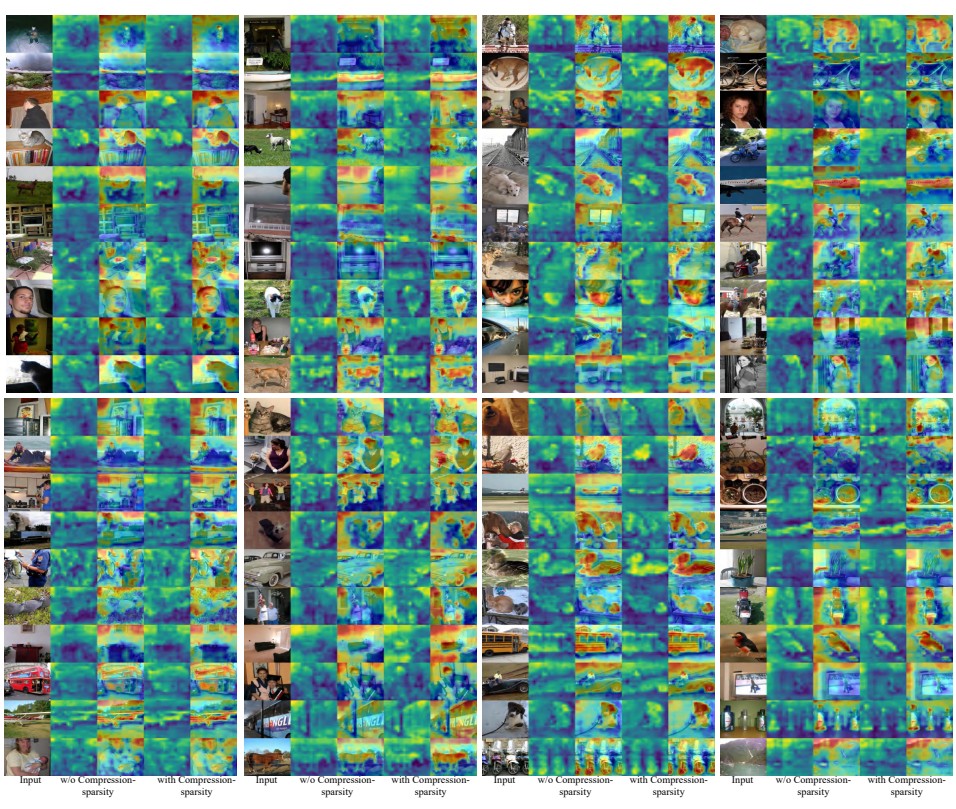

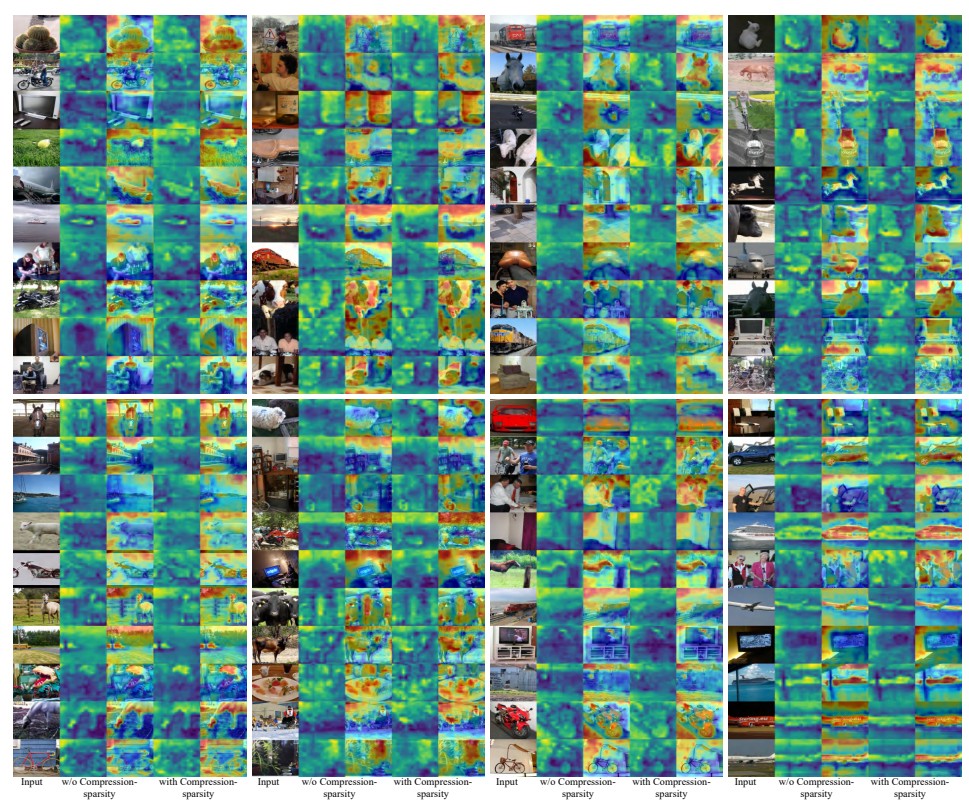

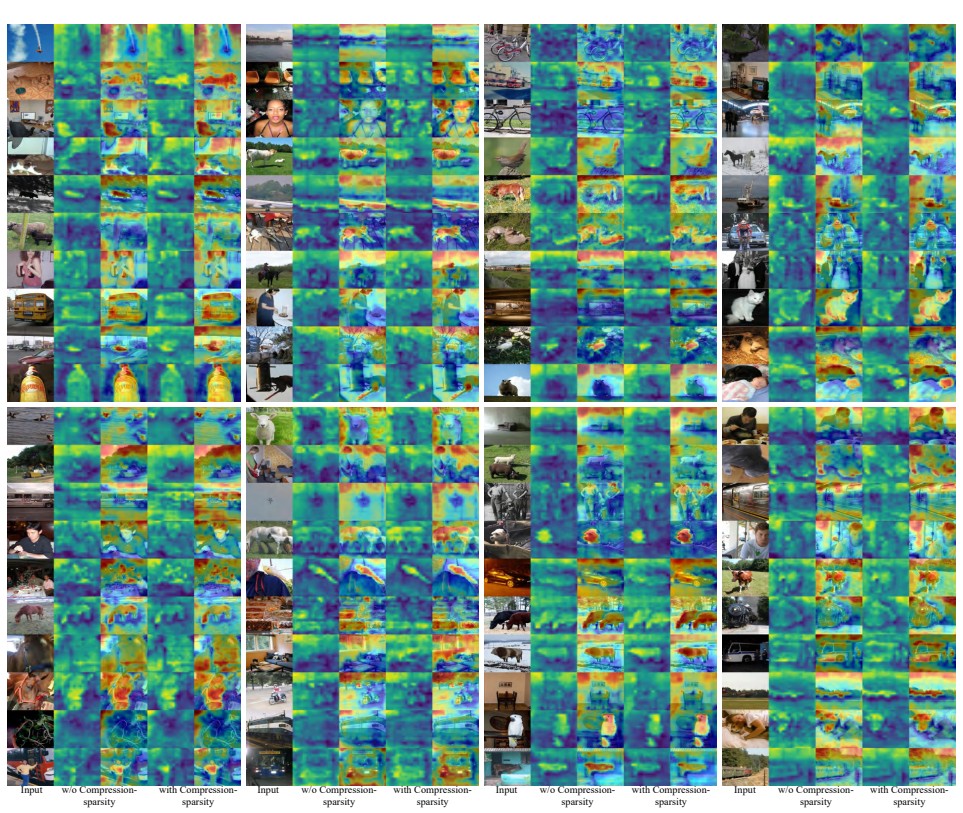

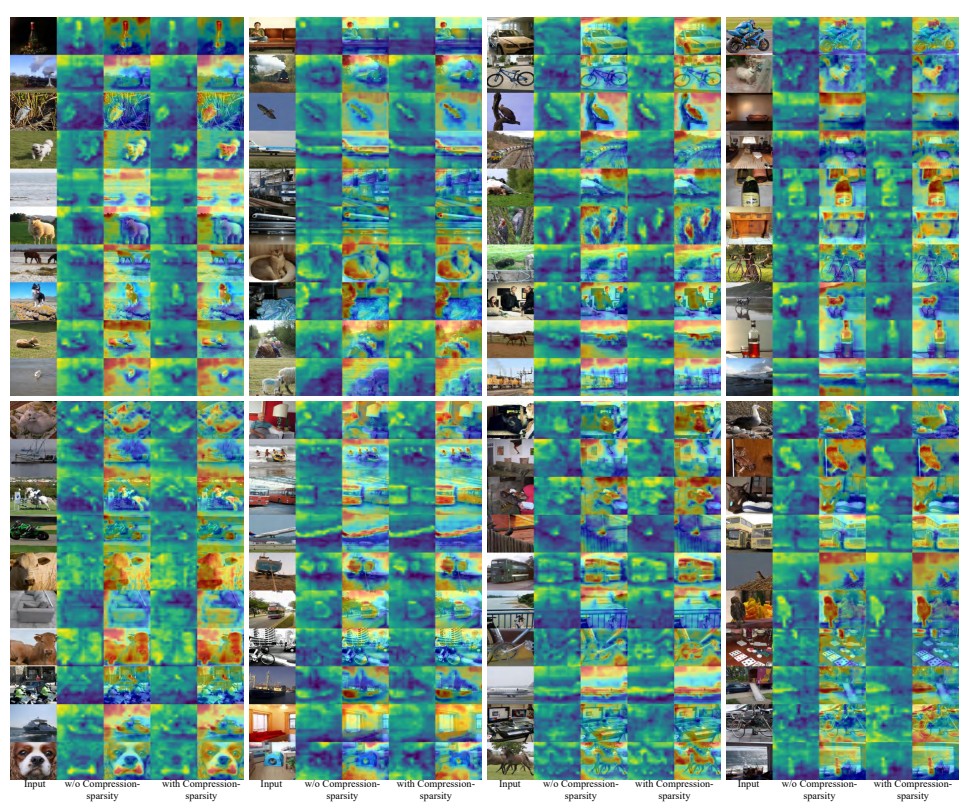

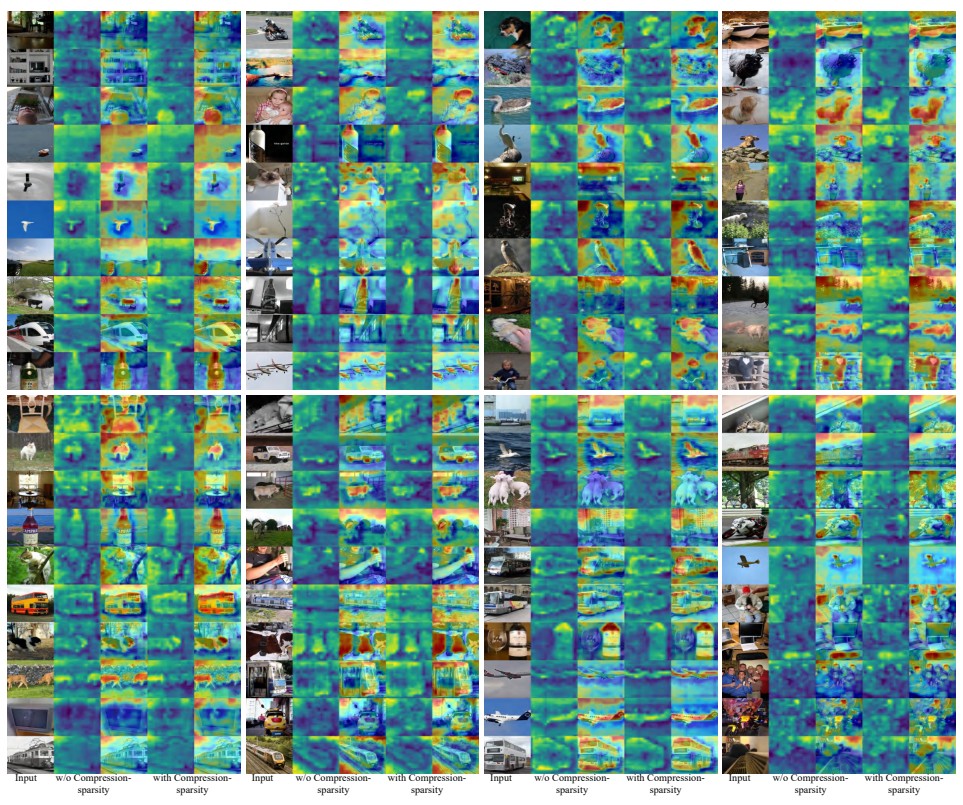

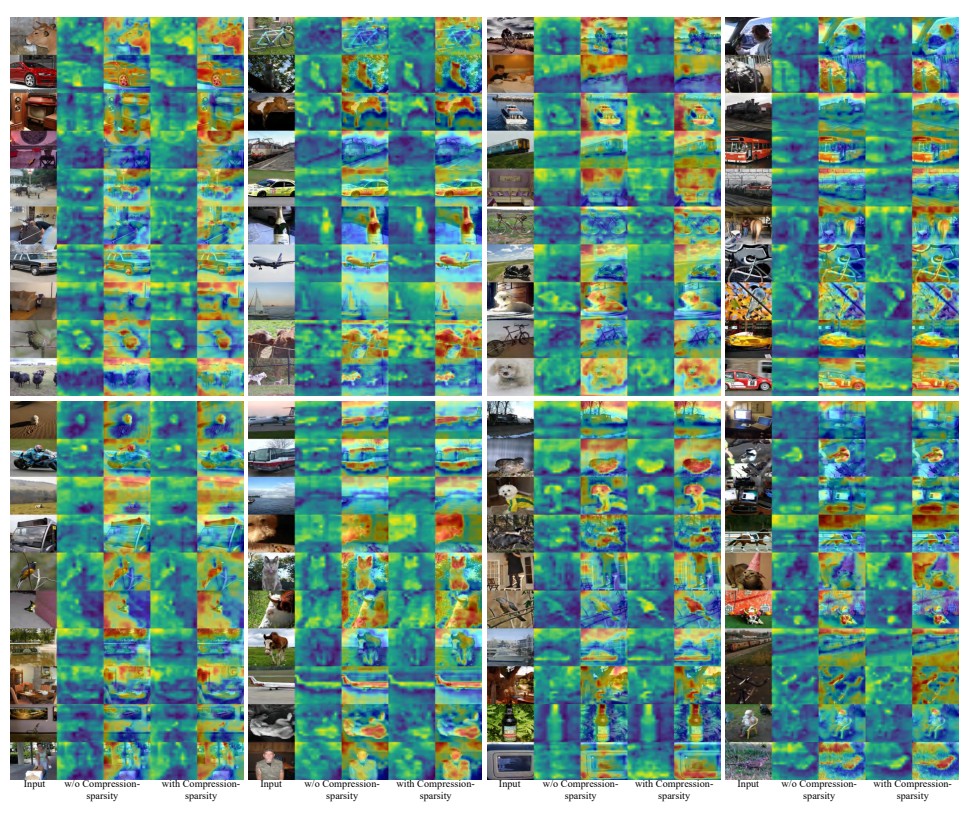

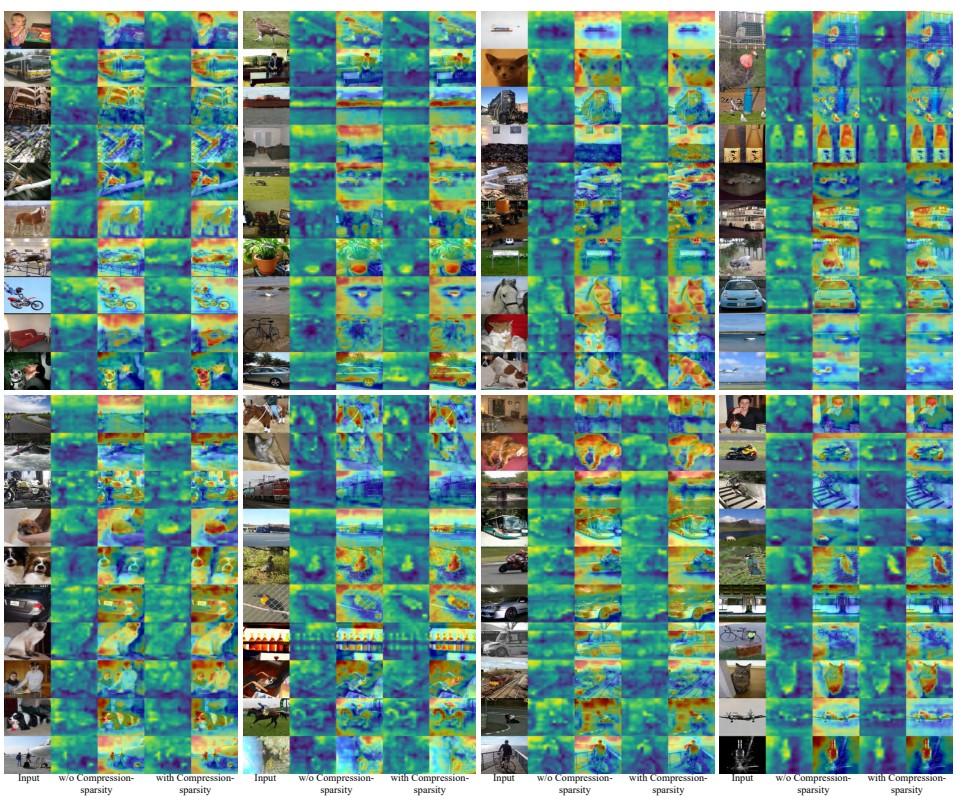

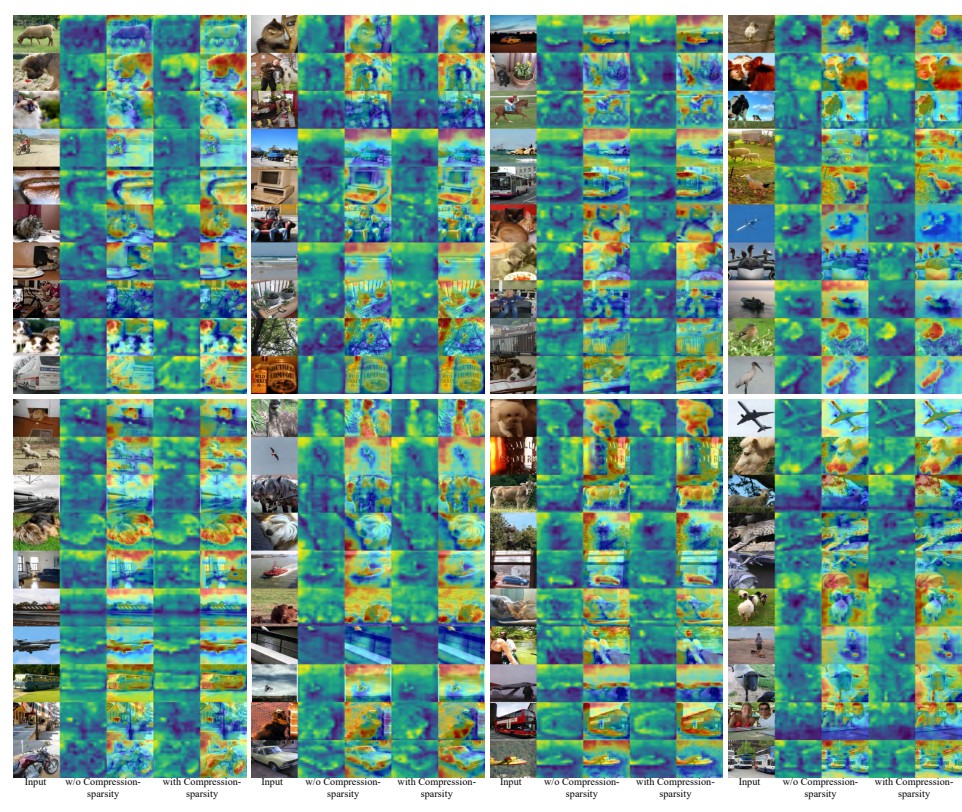

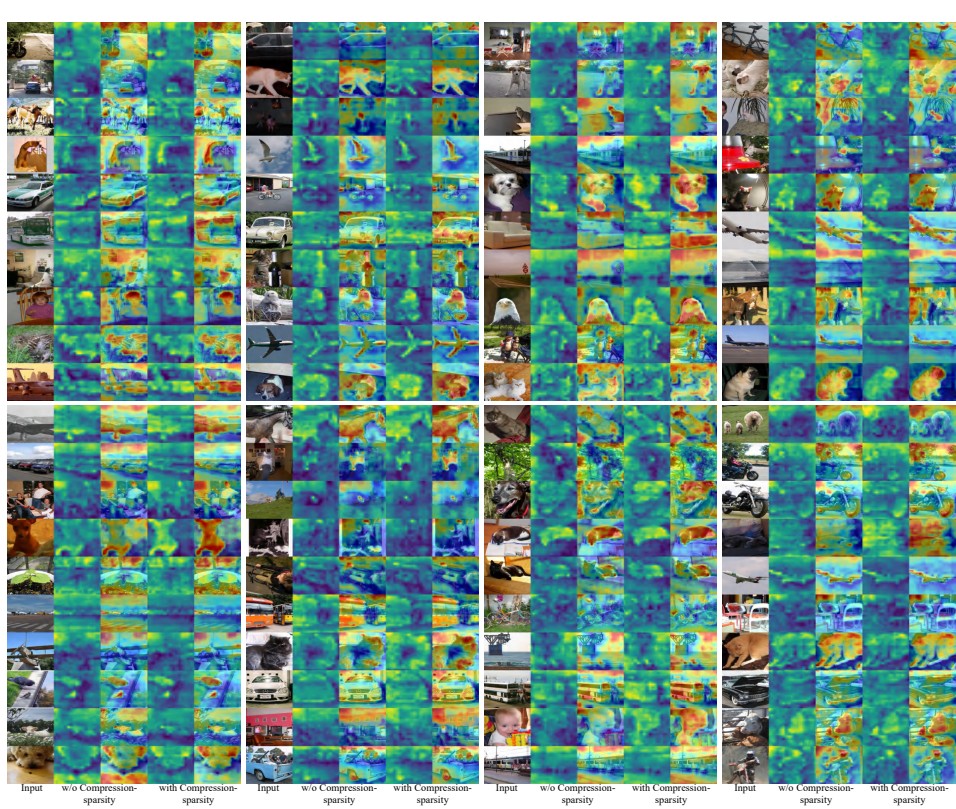

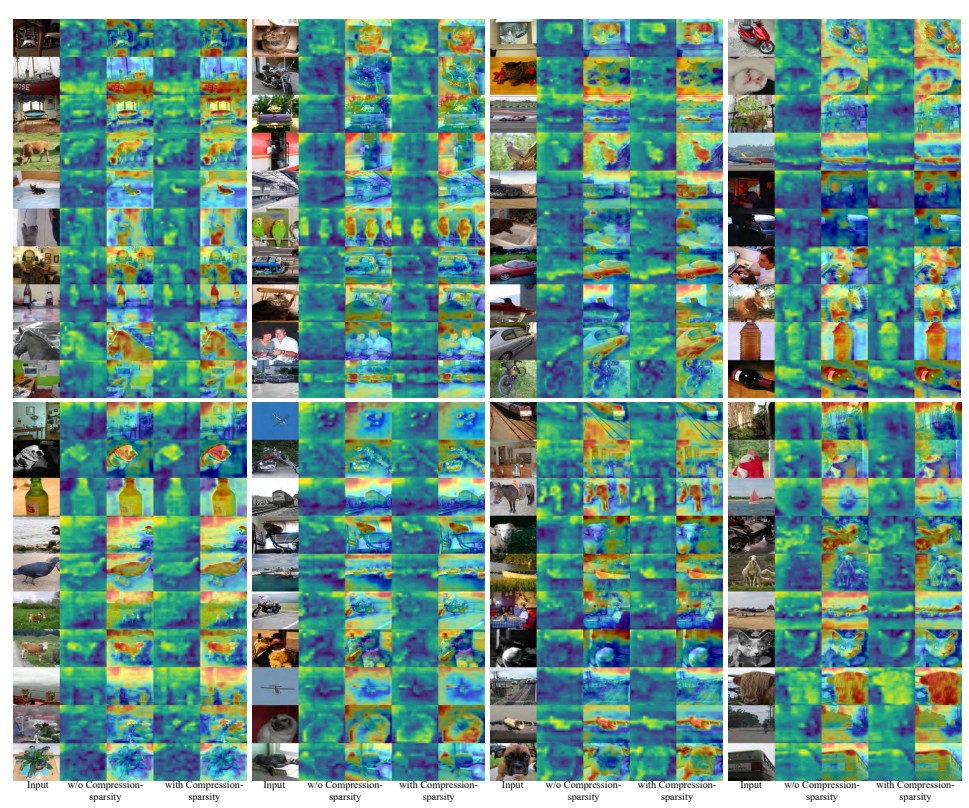

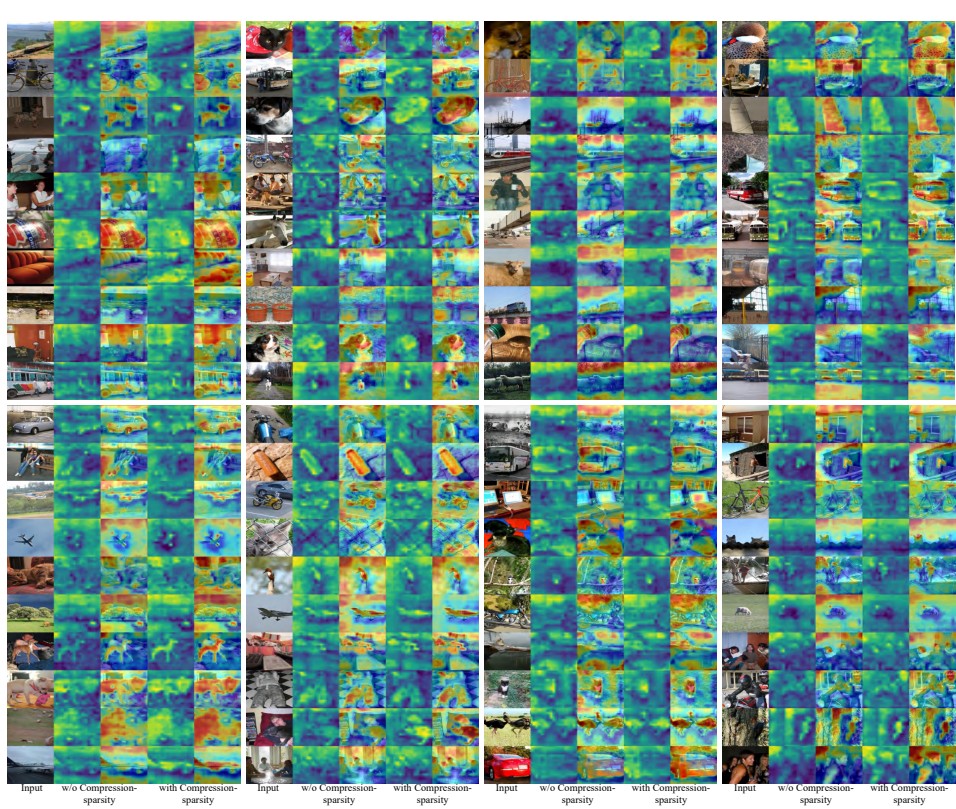

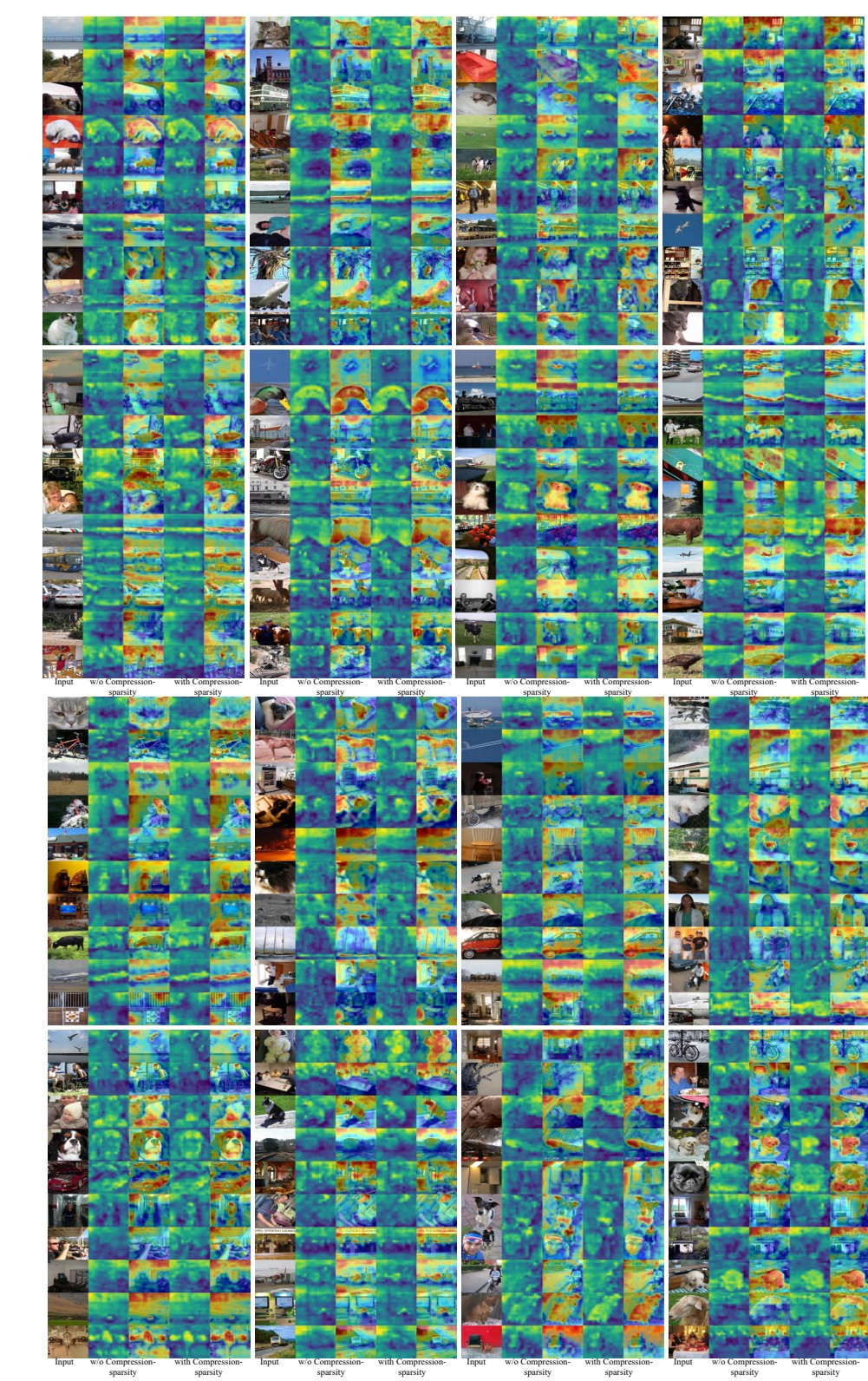

Figure 8: Feature response maps with compression-sparsity methods (columns four and five) and characteristics of maps without compression-sparsity methods (columns two and three). It can be seen that after incorporating the compression-sparsity principle, the feature responses of most data become richer and stronger, which greatly facilitates the acquisition of discriminative features.

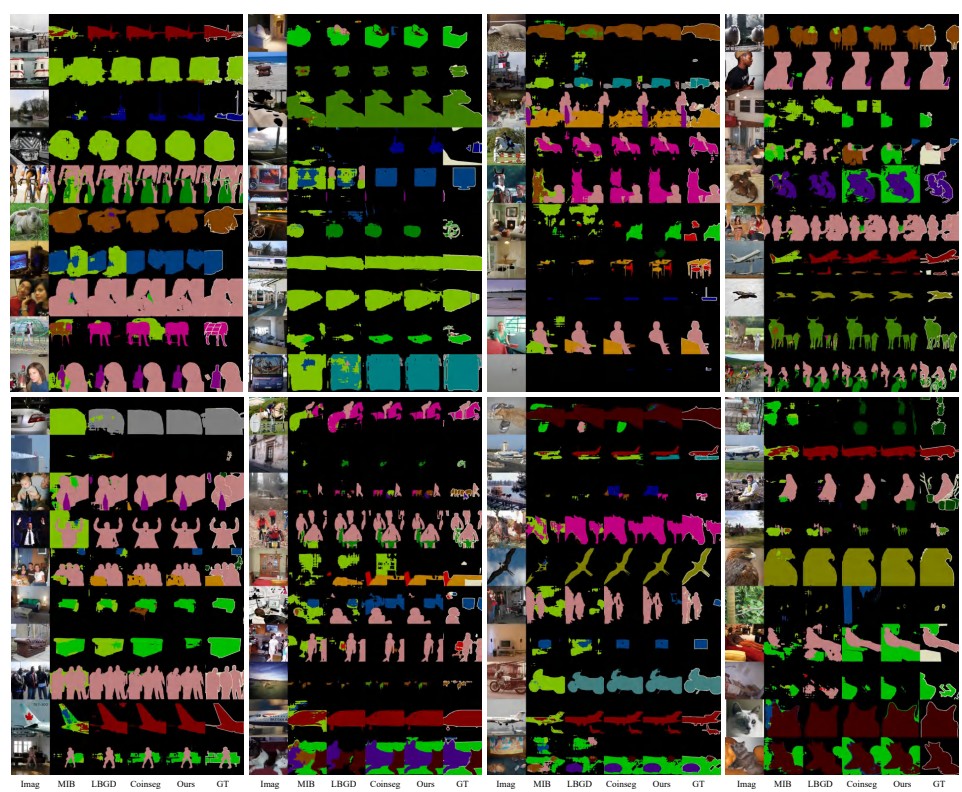

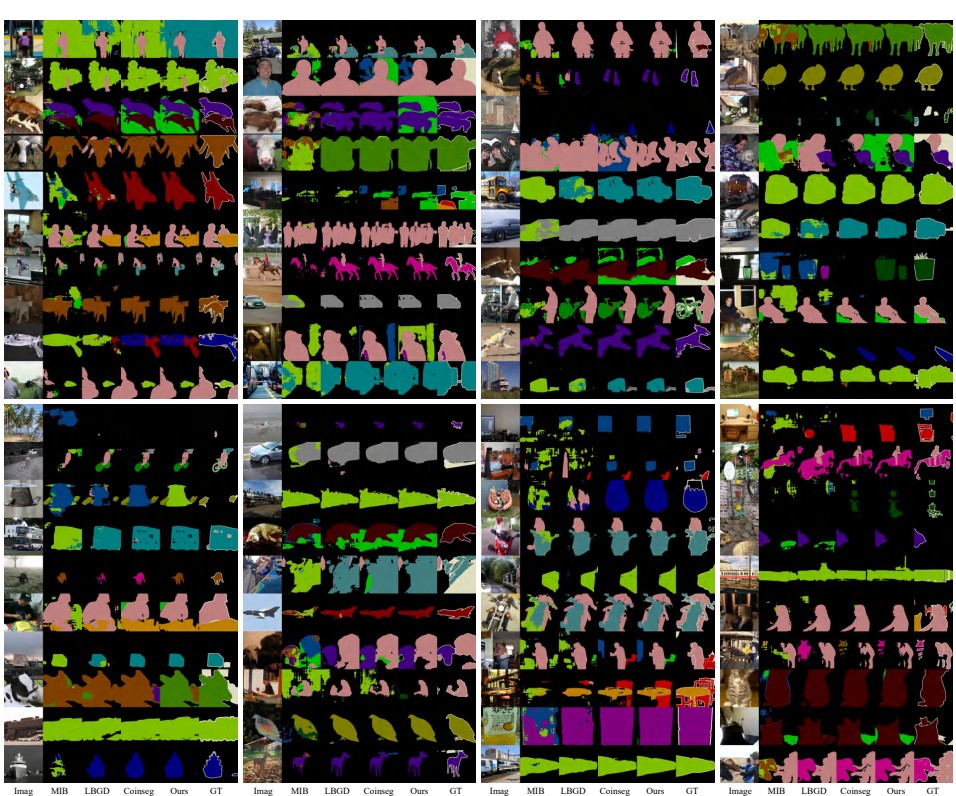

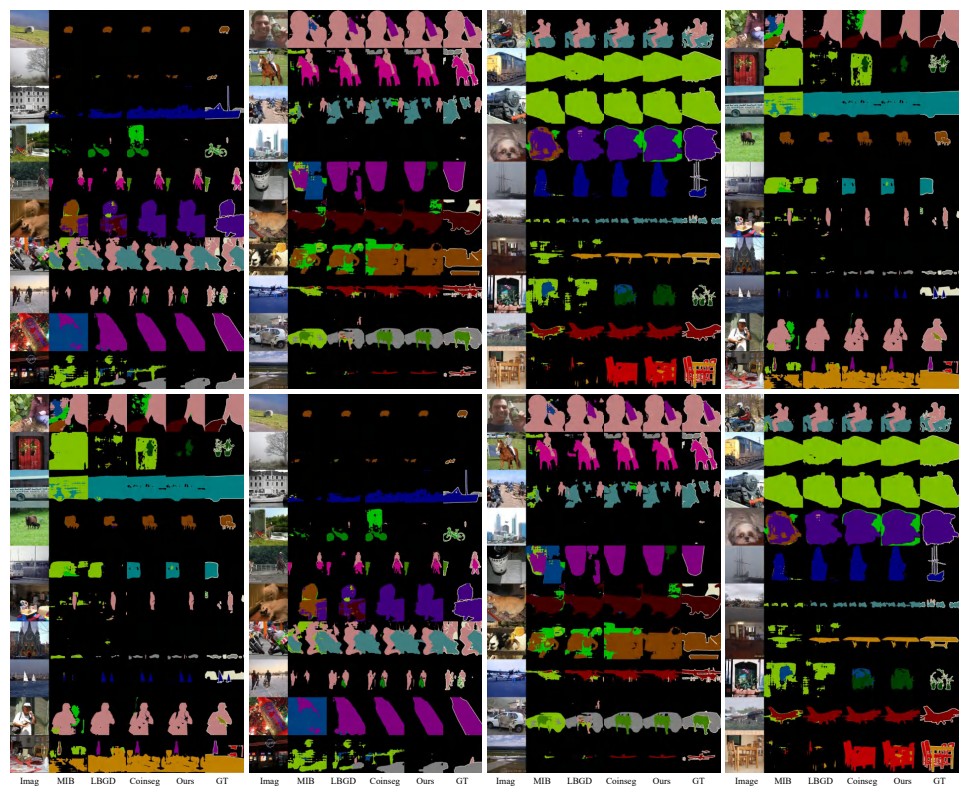

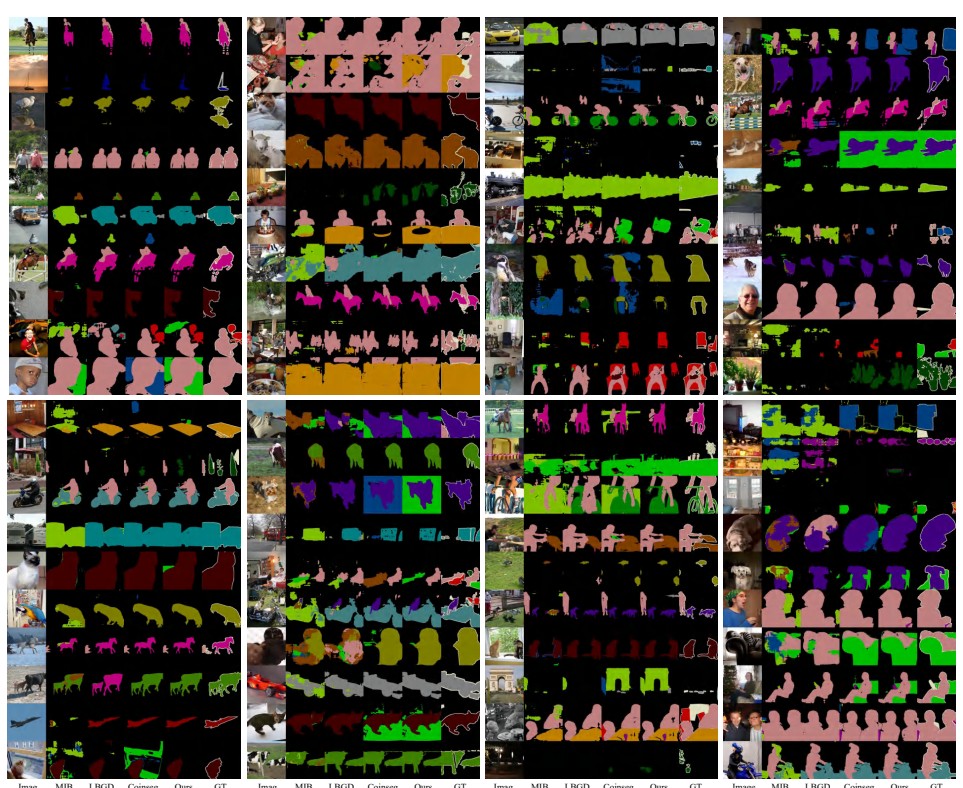

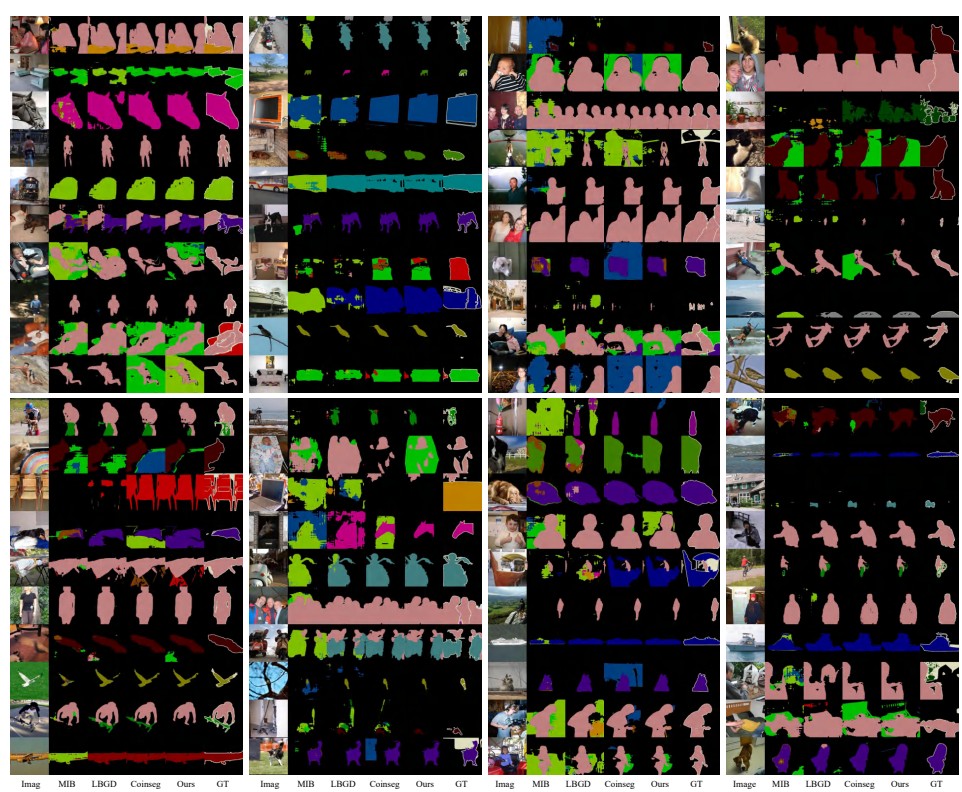

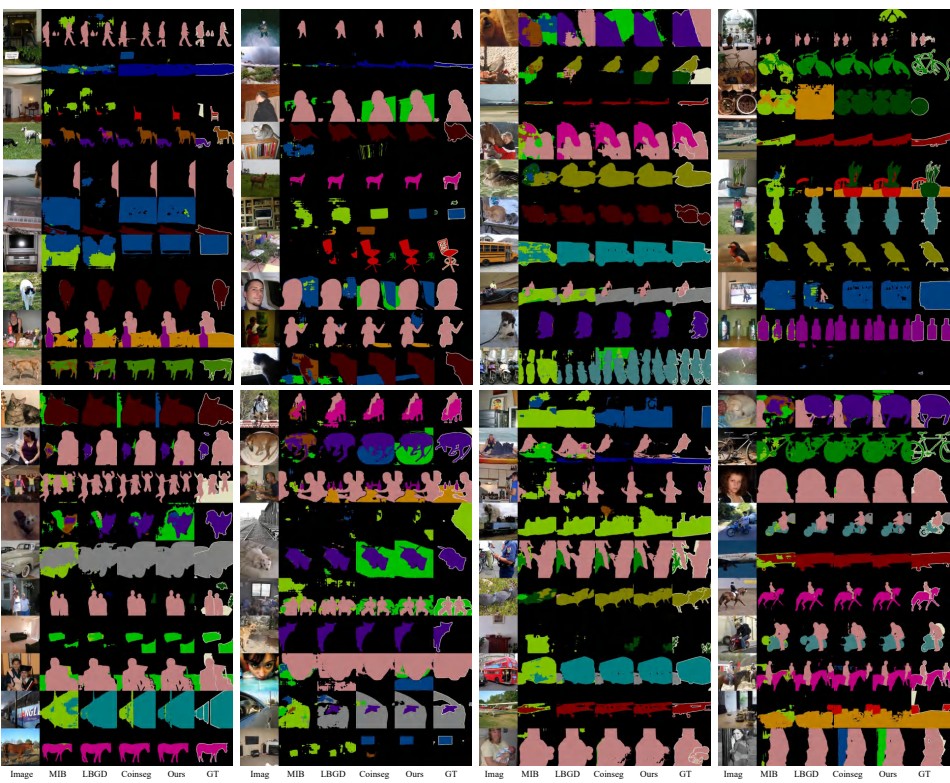

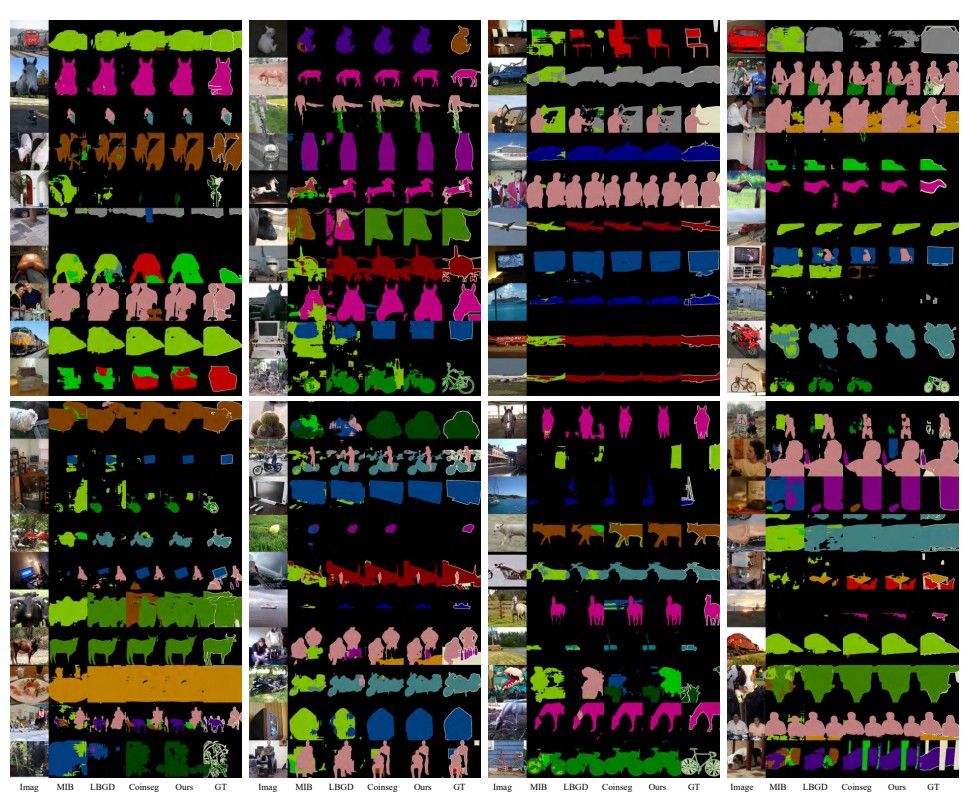

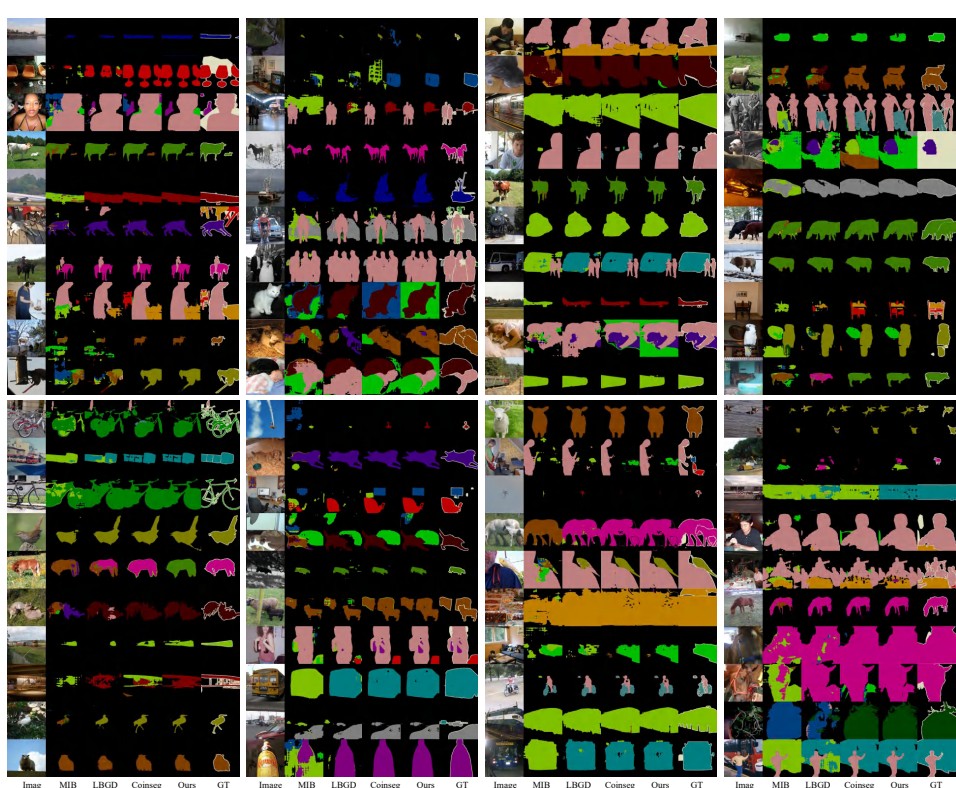

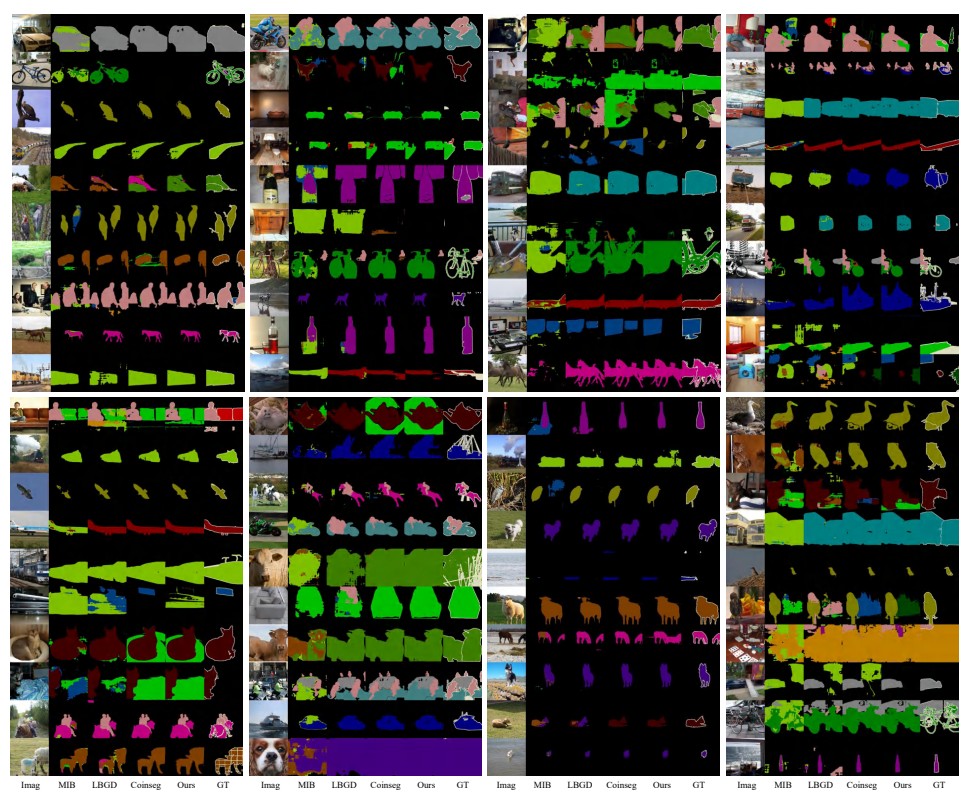

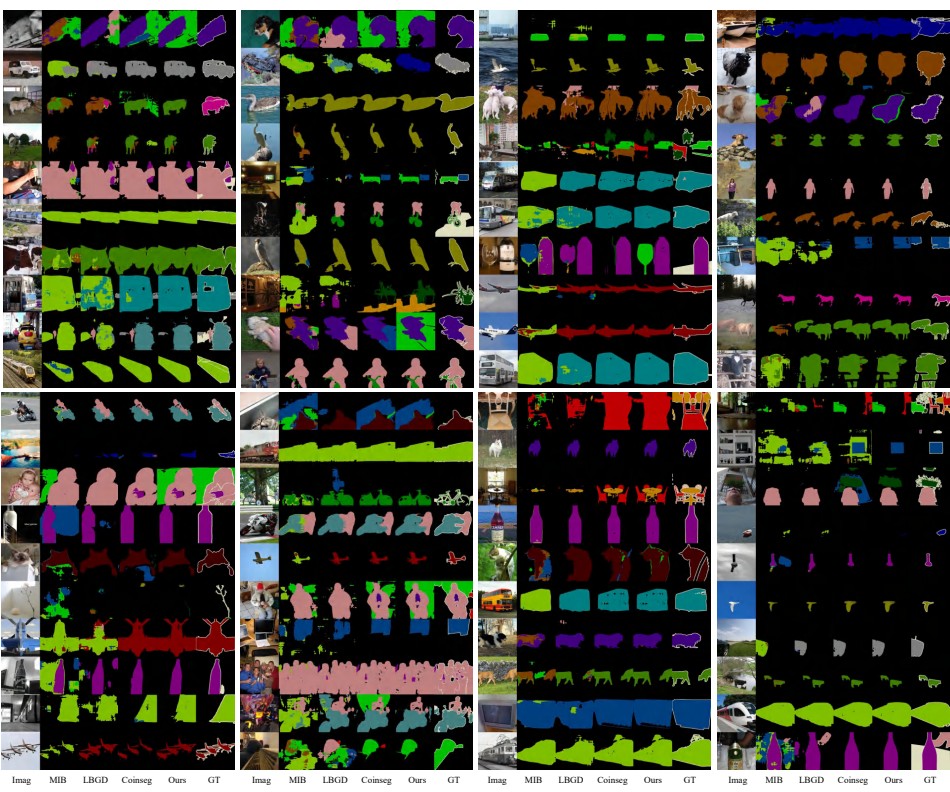

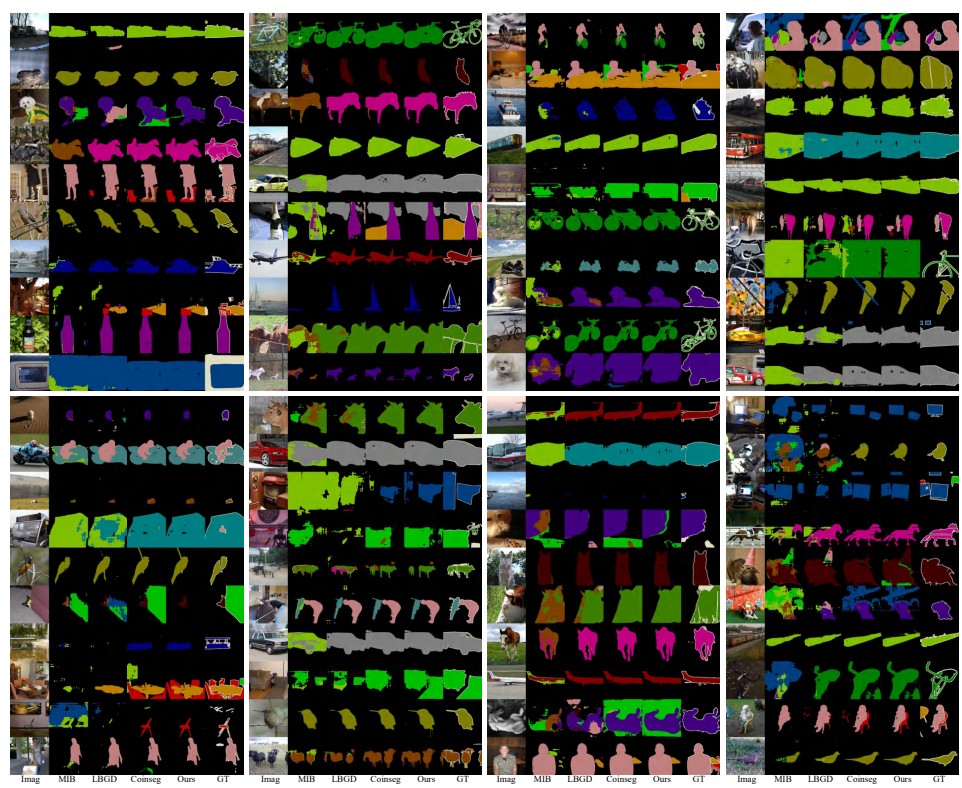

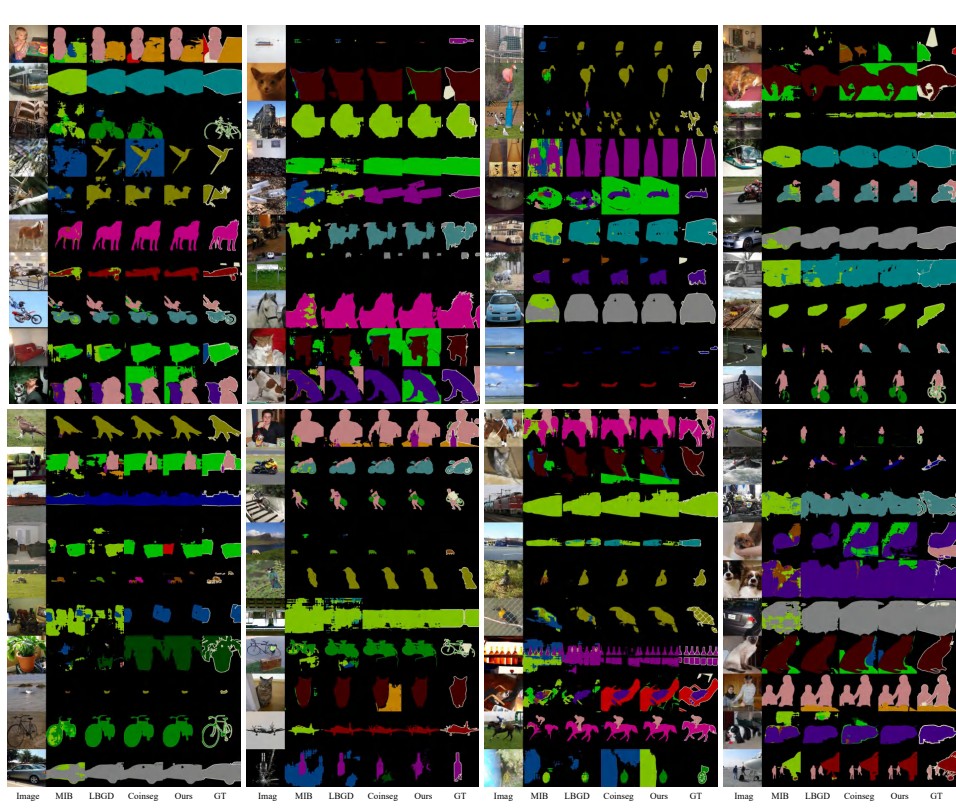

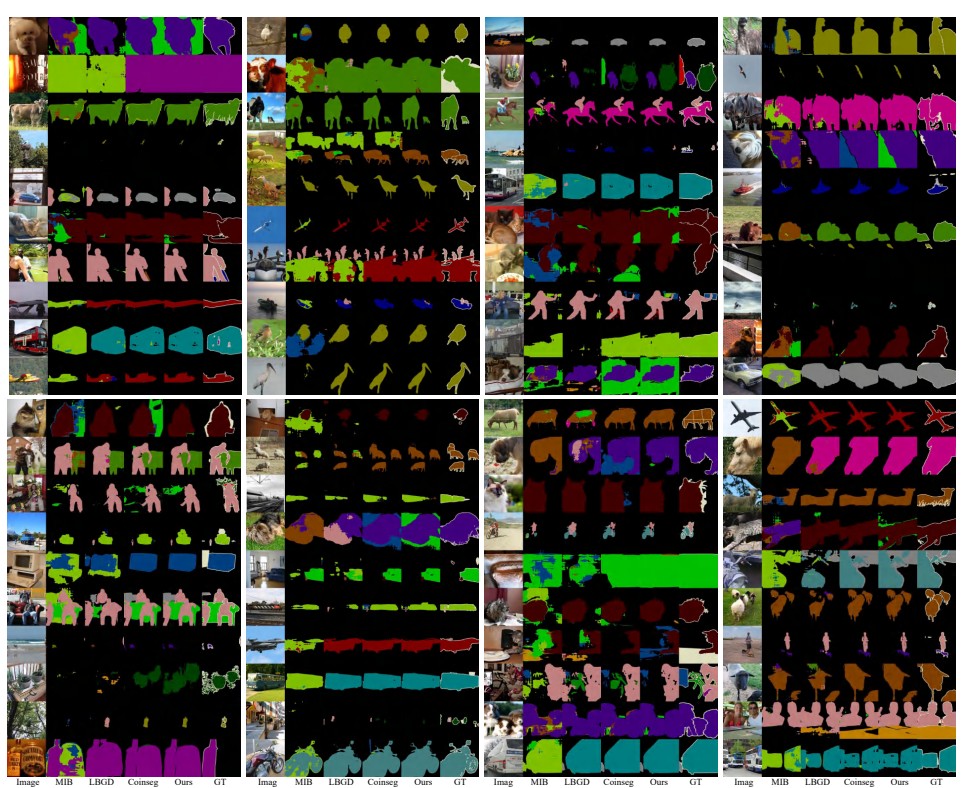

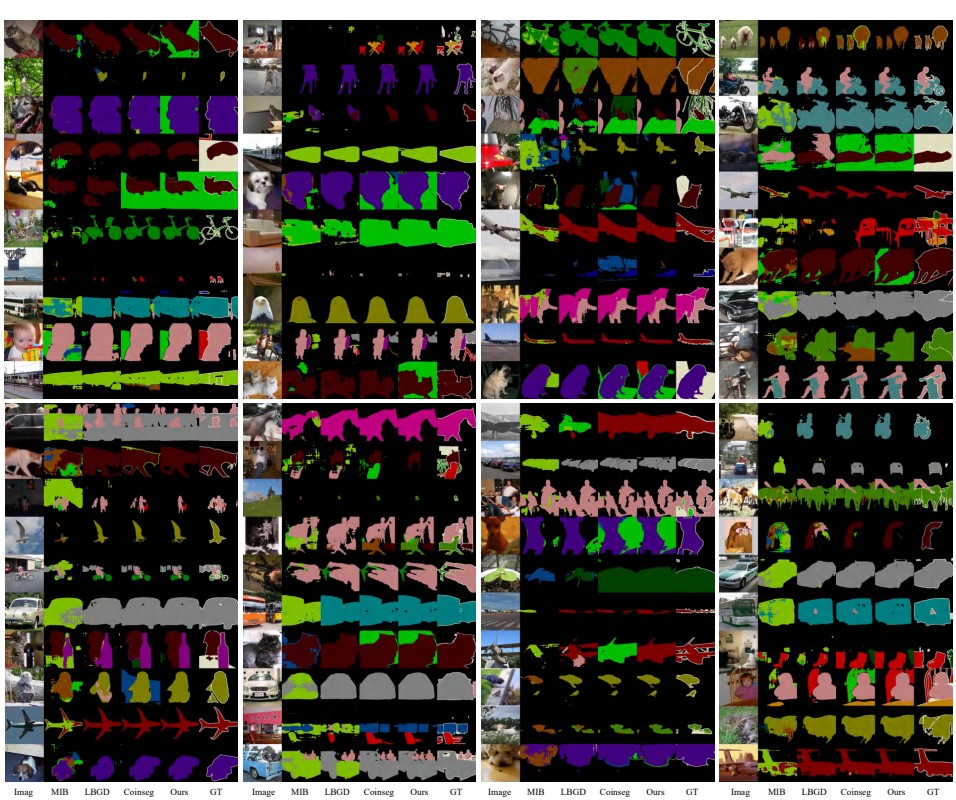

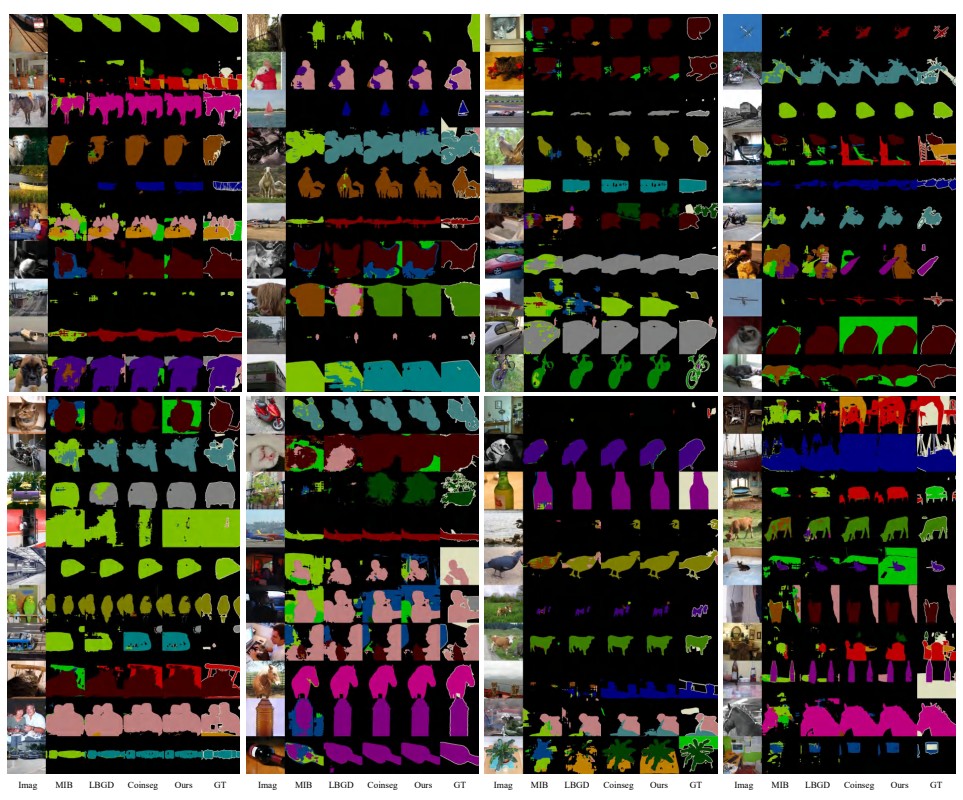

Imag MIB LBGD Coinseg Ours GT     Imag MIB LBGD Coinseg Ours GT     Imag MIB LBGD Coinseg Ours GT     Imag MIB LBGD Coinseg Ours GT

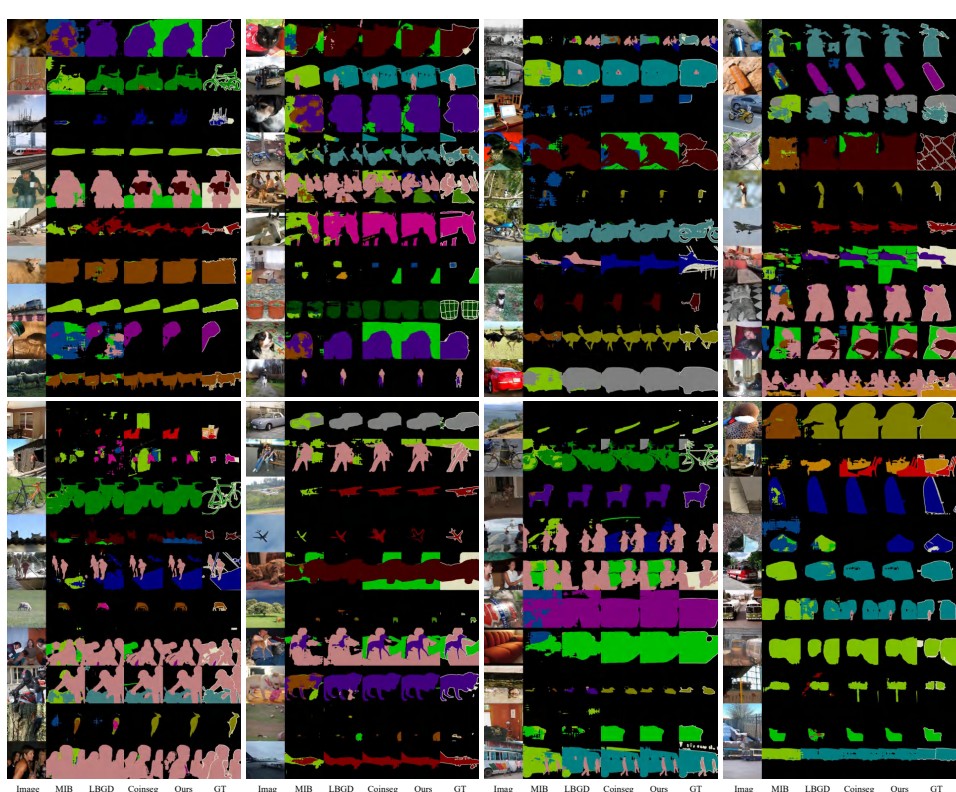

Image MIB LBGD Coinseg Ours GT     Imag MIB LBGD Coinseg Ours GT     Imag MIB LBGD Coinseg Ours GT     Imag MIB LBGD Coinseg Ours GT

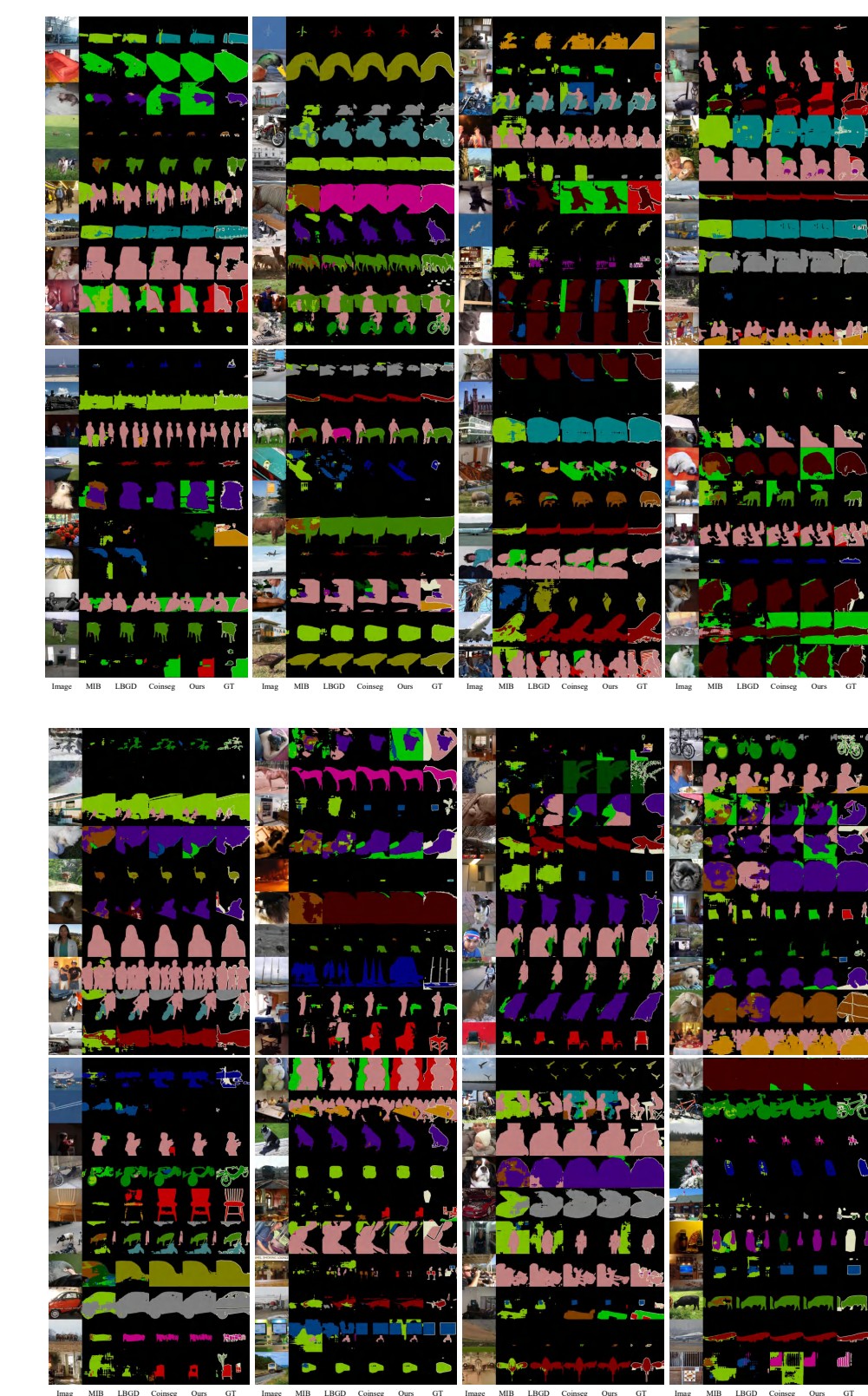

Figure 9: More comparisons with recent methods on the 15-1 testing dataset. From the results, it can be seen that our method is able to maintain good segmentation of old categories on most data and achieve effective learning of new categories.

