# OpenReview forum: "Rethinking the Influence of Distribution Adjustment in Incremental Segmentation"
_ICLR.cc/2025/Conference — ICLR 2025 Conference Withdrawn Submission_

### Official Review · Reviewer_SfE5 · 2024-10-26

**Soundness:** 3
**Presentation:** 4
**Contribution:** 2
**Rating:** 5
**Confidence:** 3

**Summary:**

The paper uses mathematical structure analysis to allocate more space for new knowledge in incremental segmentation learning, achieving dynamic adaptation of both new and old knowledge in a unified space. It significantly surpasses existing methods in the mIoU evaluation metric.

**Strengths:**

1. This paper explores how to dynamically and synchronously adjust the representations of new and old knowledge in the feature space during incremental learning. The proposed method is interesting which leavrages incremental segmentation learning method based on the compression-sparsity principle.

2. The proposed method dynamically adjusts the feature distributions of new and old knowledge to address the balance between stability and flexibility, and the effectiveness of the results is verified on mainstream datasets.

3. The writing is good and easy to follow.

**Weaknesses:**

1. The proposed method is compared with the CoinSeg swinB model. Some metrics are close to CoinSeg, while in certain cases, they are lower. Does this indicate that the effectiveness of the proposed method is insufficient?

2. The concept of plasticity should be briefly introduced when it first appears at abstract to facilitate reading.

3. In Section 3.2, the mathematical analysis of the compression-sparsity principle lacks necessary citations, such as the application of Bayes' theorem in incremental learning.

[1] Thrun, S., & Pratt, L. (1998). Learning to learn. Springer Science & Business Media
[2] Ghahramani, Z., & Beal, M. J. (2001). Propagation algorithms for variational Bayesian learning. Advances in Neural Information Processing Systems, 13, 507-513.

4. How are the compression and sparsity ratios, α and β, chosen, and are they sensitive to different datasets?

5. It is recommended to discuss the applicability of this method in real-world applications in the Limitations or Future Work section.

**Questions:**

The paper lists five incremental configurations for learning. What is the difficulty ranking among them, and how do they rank in terms of importance?

---

### Official Review · Reviewer_qxwJ · 2024-11-01

**Soundness:** 3
**Presentation:** 2
**Contribution:** 3
**Rating:** 5
**Confidence:** 3

**Summary:**

This paper proposes the Compression-Sparsity principle to dynamically adjust feature distributions. In the process of incremental learning, this principle effectively balances the stability and plasticity of the model through the compression and sparsity of the feature space. This method not only retains knowledge of old categories when introducing new ones but also mitigates the phenomenon of catastrophic forgetting. By integrating knowledge distillation with sparsity, Compression-Sparsity-based Incremental Segmentation Learning compresses the knowledge structure within complex networks while preserving inter-class discriminative features. This approach optimizes the model's feature distribution and minimizes overlap in the feature space, providing ample capacity for learning new categories.

**Strengths:**

The paper conducts an in-depth analysis from a probabilistic perspective. The experimental evaluation looks comprehensive. The supplementary material is very exhaustive.

**Weaknesses:**

The explanations of some key steps and formulas are not detailed enough, particularly in the mathematical derivations, which makes it difficult for the reader to fully understand this method.

**Questions:**

1. How does the empirical Fisher information matrix F(θ*) influence the optimization of the prior distribution in the current dataset?
2. How does feature contraction on the original spatial distribution help neural networks in searching for optimal parameters?
3. How do the compression and sparsity of feature space distribution among different classes enhance log P(X₂|θ) and log P(θ|X₁)?
4. Furthermore, I believe that the problem of catastrophic forgetting in language models primarily occurs in fine-tuning settings. Please provide a more detailed explanation of how catastrophic forgetting manifests in semantic segmentation.

---

### Official Review · Reviewer_aWzx · 2024-11-03

**Soundness:** 2
**Presentation:** 1
**Contribution:** 2
**Rating:** 3
**Confidence:** 5

**Summary:**

The paper presents a new approach to incremental segmentation that enhances plasticity and minimizes the overlap between old and new knowledge distributions in feature space. Through a mathematical framework, the authors introduce a compression-sparsity principle, wherein old knowledge representations are compressed and sparsified to allocate more feature space for new knowledge. The approach is implemented through Gaussian mixture models and achieves a balance between stability and plasticity, addressing challenges such as catastrophic forgetting.

**Strengths:**

1. The compression-sparsity approach is well-motivated. Promoting sparse and separated feature distributions provides a compelling way to prevent knowledge interference.
2. The paper provides some mathematical foundations for the proposed approach, demonstrating the benefits of this feature space adjustment through probabilistic and distribution-based analyses.

**Weaknesses:**

1. The implementation part is not clearly illustrated with confusing notations. For instance, P is used to represent peak points in the feature space while also used to denote the prediction results in Eq. 10.

2. The rationale behind Eq. 8 is not well elaborated as well. I don't see a strong correlation between Eq. 8 and the theoretical analysis in Section 3.2.

3. This paper missed the latest SOTA method [1] in CSS.

4. I suggest the authors pay more attention to simplicity and clarity, rather than wrapping up with "mathematical theories".

5. The experimental part is incomplete, missing the most important benchmark ADE20k. I recommend the authors make the experiments more comprehensive, address the weakness mentioned above, and resubmit the manuscript to another conference for review.

[1] Kim, Beomyoung, Joonsang Yu, and Sung Ju Hwang. "ECLIPSE: Efficient Continual Learning in Panoptic Segmentation with Visual Prompt Tuning." Proceedings of the IEEE/CVF Conference on Computer Vision and Pattern Recognition. 2024.

**Questions:**

Eq. 8 should be elaborated with details. How is each part implemented and how do they contribute to the feature sparsity?

---

### Official Review · Reviewer_qPhV · 2024-11-04

**Soundness:** 3
**Presentation:** 3
**Contribution:** 3
**Rating:** 6
**Confidence:** 4

**Summary:**

Considering stability and plasticity, this paper argues that compressing the feature subspace and promoting sparse distribution is beneficial in allocating more space for new knowledge in incremental segmentation learning. And then proposes compression-sparsity loss to conduct effective knowledge transfer. The authors conduct experiments and show improvement.

**Strengths:**

From the perspective of compression and sparsity, the paper is well-motived and proposes a loss function that can be used in many baselines.

**Weaknesses:**

As a distillation-related loss function, this paper lacks of direct comparison with MiB in the method. Some experiments on ADE20K are missed.

**Questions:**

- In Table 2, as a commonly used benchmark, the experiments on the 100-50, 100-10, and 50-50 settings are missed.
- As a loss function related to distillation, how about the performance on different baseline models in Table 1?
- I wonder is there any way to measure the sparsity and compression of the features trained by $L_{cs}$?
- In Table 3, the highest performance is from KD + C rather than KD + C + S. Is there any insight or explanation?
- In Table 4, why does the weighted approach outperform the attention approach by a large margin?
- It seems that the C-S principle can also be applied to the incremental classification task. Could you please report the results of the classification task to verify the effectiveness?
- Is the method compatible with the pseudo-labels-based method?

---

### Note · Authors · 2024-11-14

I have read and agree with the venue's withdrawal policy on behalf of myself and my co-authors.